# iGraphMix: Input Graph Mixup Method for Node Classification

**Jongwon Jeong**[1]***Hoyeop Lee**[2]**, Hyui Geon Yoon**[2]**, Beomyoung Lee**[2]**, Junhee Heo**[2]**,
Geonsoo Kim**[2]**, Jin Seon Kim**[2]
KRAFTON[1], NCSOFT Co.[2]

## Abstract

Recently, Input Mixup, which augments virtual samples by interpolating input features and corresponding labels, is one of the promising methods to alleviate the over-fitting problem on various domains including image classification and natural language processing because of its ability to generate a variety of virtual samples, and ease of usability and versatility. However, designing Input Mixup for the node classification is still challenging due to the irregularity issue that each node contains a different number of neighboring nodes for input and the alignment issue that how to align and interpolate two sets of neighboring nodes is not well-defined when two nodes are interpolated. To address the issues, this paper proposes a novel Mixup method, called iGraphMix, tailored to node classification. Our method generates virtual nodes and their edges by interpolating input features and labels, and attaching sampled neighboring nodes. The virtual graphs generated by iGraphMix serve as inputs for graph neural networks (GNNs) training, thereby facilitating its easy application to various GNNs and enabling effective combination with other augmentation methods. We mathematically prove that training GNNs with iGraphMix leads to better generalization performance compared to that without augmentation, and our experiments support the theoretical findings.

## 1 Introduction

*Mixup* (Zhang et al., 2018) is one of the effective data augmentation methods that help prevent the over-fitting problem of neural networks, particularly caused by a lack of labeled data in training. Mixup augments data by randomly selecting two samples from training sets and linearly interpolating samples and corresponding labels. Previous Mixup methods can be categorized into Input Mixup (Zhang et al., 2018; Yun et al., 2019), which directly interpolates input features and corresponding labels, and Manifold Mixup (Guo et al., 2019; Verma et al., 2019), which accesses the intermediate layer of neural networks to get hidden representations and interpolates them and corresponding labels.

Recent studies in Mixup have been primarily focused on Input Mixup in various domains including image classification and natural language processing, due to its two advantages. First, Input Mixup allows for the generation of a wider range of samples compared to Manifold Mixup. This is because Input Mixup can generate a variety of virtual samples by employing several interpolation methods via input characteristics of each domain, such as patch incorporation from two images (Yun et al., 2019) and substructure merging from two sentences (Zhang et al., 2022). In contrast, the mixing hidden representation has limitations on the diversity of interpolation because the embedding model projects an input space into an ambiguous space (Yoon et al., 2021). Second, Input Mixup has ease of usability and versatility. It does not need to access the intermediate layers, thus eliminating the need for modifications to the models and making it applicable to any model architecture (Yoon et al., 2021). In addition, it can be versatile with other augmentation techniques, enabling further extension studies (Berthelot et al., 2019).

Due to the dissimilar input structure of graph neural networks (GNNs), *irregularity* and *alignment* issues may arise in applying Input Mixup straightforwardly to the node classification (Wang et al., 2021). These issues render the application of Input Mixup challenging in the context of GNNs for

---

*This work was done at NCSOFT Co. Correspondence to: Jongwon Jeong<`jwjeong@krafton.com`>.

node classification. GNNs receive not only the target nodes but also their corresponding neighboring nodes for the inputs. Then, GNNs employ a message-passing process to propagate information from neighboring nodes to the target nodes, followed by an aggregation process that combines the information of the neighboring and target nodes for prediction. This process introduces the irregularity and alignment issues that make it difficult to design Input Mixup for GNNs. The irregularity issue arises from the input of various numbers of neighboring nodes among target nodes, resulting in different input sizes between two target nodes in GNNs for interpolation. The alignment issue stems from the lack of inherent ordering among two neighboring node sets, leading to the challenge of how the nodes are applied to the message-passing and the aggregation process even though the inputs are interpolated. As a result, previous works (Verma et al., 2021; Wang et al., 2021) on Mixup for node classification have relied on Manifold Mixup. They prevent these issues but sacrifice the advantages of Input Mixup due to accessing hidden representations.

To cope with the two issues of designing Input Mixup to node classification in GNNs while sustaining the two advantages, we propose a simple yet effective Input Mixup method, called iGraphMix. The proposed method interpolates two nodes' input features, labels, and edges as follows. First, to generate a virtual feature and label for a virtual training node, iGraphMix interpolates nodes' input features and associated labels, as the original Input Mixup does. Second, to construct virtual edges for a virtual node, iGraphMix randomly selects the neighboring nodes from each node. Then, our method attaches all selected neighboring nodes to generate the virtual edges. This attaching process for the virtual edges enables iGraphMix to avoid the irregularity issue occurring at interpolating the different edges of two nodes. Our method also helps to prevent the alignment issue by employing the message-passing and aggregation processes with the virtual edges. Furthermore, randomly selecting neighboring nodes in our method enables the generation of diverse virtual nodes without model modification, thus retaining the two advantages of Input Mixup.

Our study provides three main contributions. First, to the best of our knowledge, iGraphMix is the first Input Mixup method designed specifically for node classification in the graph domain, addressing the irregularity and alignment issues of designing Input Mixup for GNNs. We believe that this work can serve as foundation research for Input Mixup on graphs, similar to recent developments in various domains such as image classification (Yun et al., 2019) and natural language processing (Kong et al., 2022). Second, we provide theoretical analysis and experimental validation of how our method reduces the generalization gap. The results emphasize the importance and effectiveness of iGraphMix to regularize GNNs. Last, we experimentally confirm that training GNNs with the proposed method outperforms GNNs trained without augmentations by an average of 2.84% in terms of the micro-F1 score across five benchmark datasets using three GNN models. We conclude that iGraphMix is an effective method to prevent the over-fitting problem and improve the performance of GNNs.

## 2 RELATED WORKS

**Mixup**   Mixup has been recently highlighted due to its ability to mitigate the distorted interrelationship problem arising in input data augmentation methods (Shorten & Khoshgoftaar, 2019; Naveed et al., 2021). Since they perturb only inputs (DeVries & Taylor, 2017; Singh et al., 2018; Edunov et al., 2018; Kobayashi, 2018), neural networks may learn the distorted interrelationship between augmented inputs and their corresponding labels (Zhang et al., 2018). Mixup can alleviate this problem by interpolating samples and their labels. While Mixup can be categorized into Input Mixup (Zhang et al., 2018) and Manifold Mixup (Verma et al., 2019), recent studies have mainly focused on Input Mixup due to the ability to generate diverse virtual nodes and usability and versatility. Therefore, several Input Mixup methods were proposed to generate a variety of virtual samples depending on the characteristics of input structures in many domains (Kim et al., 2020; Qin et al., 2020; Liu et al., 2022b; Kong et al., 2022; Zhang et al., 2022). For instance, in the image classification, Yun et al. (2019) extracted patches from two images and replaced each other to get augmented images. In natural language processing, Yoon et al. (2021) mixed two sentences by replacing a span of words, which is a meaningful set of words in one sentence, with that in others. In contrast, iGraphMix is the first Input Mixup method designed especially for node classification that considers the characteristics of the input graph structure, setting it apart from Input Mixup methods in other domains.

**Data Augmentation for Node Classification**   Data augmentation is an effective regularization technique to alleviate the over-fitting problem that often occurs in node classification due to a lack of

labeled nodes (Zhao et al., 2022). Several works tried to augment input graph structures to alleviate this problem. For instance, DropEdge (Rong et al., 2020) and its variant (Gao et al., 2021) randomly drop edges in the training graphs. DropNode (Feng et al., 2020; You et al., 2020) arbitrarily masks nodes and connected edges to augment training graphs. DropMessage (Fang et al., 2023) randomly permutes passing messages from neighboring nodes. Other methods (Wang et al., 2020; Zhao et al., 2021; Park et al., 2021; Liu et al., 2022a) generate synthetic sub-graphs via auxiliary models or losses.

Meanwhile, some methods (Verma et al., 2021; Wang et al., 2021) were proposed to design Manifold Mixup methods to augment inputs and corresponding labels for node classification due to alleviating the distorted interrelationship problem. They relied on auxiliary techniques to access hidden representations due to the irregularity and alignment issues on the input graph. Specifically, GraphMix (Verma et al., 2021) utilizes not only GNNs but also introduces an additional fully connected neural network. Then, it applies Manifold Mixup on the nodes' hidden representation of the additional network. M-Mixup (Wang et al., 2021) augments two graph batches by permuting the order of nodes and edges in its augmentation process. Then, it interpolates the hidden representations of these two graph batches at each layer within GNNs. However, it was often challenging to generate diverse samples from those methods due to linearly interpolating hidden representation. Additionally, they lacked theoretical insights into the factors that contribute to the enhanced performance of GNNs through their methods. This paper proposes iGraphMix that directly addresses the issues for designing Input Mixup on node classification as well as preserves the two advantages that Input Mixup has. Also, we provide theoretical insight into why iGraphMix in node classification reduces the generalization gap.

## 3 PRELIMINARIES

**Notations** Let $\mathcal{G} = (\mathcal{V}, \mathcal{E})$ be an undirected graph with $n$ nodes $v_i \in \mathcal{V}$ and edges $(v_i, v_j) \in \mathcal{E}$. For ease of mathematical manipulation, we define the matrix $\boldsymbol{X} \in \mathbb{R}^{n \times d_0}$ is the node feature matrix of $n$ nodes with feature dimension $d_0$, $\boldsymbol{A} \in \mathbb{R}^{n \times n}$ is the adjacency matrix with $A_{ij} = 1$ if $(v_i, v_j) \in \mathcal{E}$ and $A_{ij} = 0$ if $(v_i, v_j) \notin \mathcal{E}$. $\boldsymbol{Y} \in \mathbb{R}^{n \times c}$ is the one-hot label matrix for nodes with given $c$ classes. Note that, $\boldsymbol{X}_v$ indicates a $v$-th row vector of matrix $\boldsymbol{X}$. It means that each node $v$ associates features $\boldsymbol{X}_v$ and corresponding label $\boldsymbol{Y}_v$.

**Graph Neural Networks** GNNs have shown remarkable success in node classification through message-passing and aggregation processes (Kipf & Welling, 2017; Veličković et al., 2018; Wu et al., 2019). The objective of training GNNs is to learn a differentiable function $f$ such that $\boldsymbol{Y} \approx \boldsymbol{Z} = f(\boldsymbol{X}, \boldsymbol{A})$. One basic structure of GNNs is a graph convolutional network (GCN). The $K$-layer GCN model $f$ is formulated as

$$
\begin{aligned}
\boldsymbol{H}^k &= \sigma\left(\bar{\boldsymbol{A}} \boldsymbol{H}^{k-1} \boldsymbol{W}^k\right), \quad \forall k \in \{1, \cdots, K-1\} \\
\boldsymbol{Z} &= \bar{\boldsymbol{A}} \boldsymbol{H}^{K-1} \boldsymbol{W}^K,
\end{aligned}
\tag{1}
$$

where $\boldsymbol{W}^k \in \mathbb{R}^{d_{k-1} \times d_k}$ and $\boldsymbol{W}^K \in \mathbb{R}^{d_{K-1} \times c}$ are the weight matrix of $k$-th layer for $k < K$ and $K$-th layer respectively, and $\bar{\boldsymbol{A}}$ be the adjacency diffusion operator. Here, we initialize $\boldsymbol{H}^0 = \boldsymbol{X}$, and $\sigma$ are the point-wise activation functions that are assumed to be Lipschitz functions with Lipschitz constant $L_\sigma$, respectively. The most popular adjacency diffusion operators are $\bar{\boldsymbol{A}} = \boldsymbol{A} + \boldsymbol{I}$ and $\bar{\boldsymbol{A}} = (\boldsymbol{D} + \boldsymbol{I})^{-\frac{1}{2}}(\boldsymbol{A} + \boldsymbol{I})(\boldsymbol{D} + \boldsymbol{I})^{-\frac{1}{2}}$ where $\boldsymbol{D}$ is a degree matrix of $\boldsymbol{A}$ (Kipf & Welling, 2017). Also, diffusion operators are formulated as attention methods so that different importance weights of edges are applied to the aggregation process (Veličković et al., 2018; Brody et al., 2022).

**Theoretical Analysis on GNNs** For the theoretical analysis, we focus GCN on the *transductive* node classification task. In transductive node classification, GNNs are trained on a subset of nodes within a specific graph and subsequently evaluated on the remaining nodes of that same graph. When training, the information of labeled nodes is backpropagated. Then, we state the generalization loss $\mathcal{L}$ and empirical loss $\hat{\mathcal{L}}$ of GCN model $f$ as follows:

$$
\mathcal{L}(f|\boldsymbol{X}, \boldsymbol{Y}, \boldsymbol{A}) = \frac{1}{n-m} \sum_{i=m+1}^{n} \ell\left(\boldsymbol{Z}_i, \boldsymbol{Y}_i\right),
\tag{2}
$$

$$
\hat{\mathcal{L}}(f|\boldsymbol{X}, \boldsymbol{Y}, \boldsymbol{A}) = \frac{1}{m} \sum_{i=1}^{m} \ell\left(\boldsymbol{Z}_i, \boldsymbol{Y}_i\right),
\tag{3}
$$

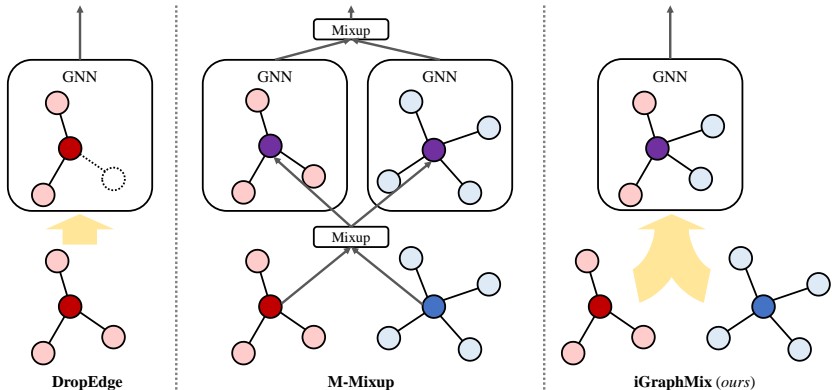

Figure 1: Illustrative comparison of DropEdge, M-Mixup, and iGraphMix. Red and blue circles indicate labeled nodes, and pink and sky-blue circles are neighboring nodes. When red and blue circles are mixed, they become purple. DropEdge randomly drops edge connection, and M-Mixup interpolates hidden representations. iGraphMix generates a virtual training node by selecting neighboring nodes and interpolating node features.

where the first $m$ nodes are labeled nodes, also called training nodes, and $\ell$ is the loss function, *e.g.*, cross-entropy loss.

Esser et al. (2021) have recently provided a theoretical understanding of GCN. They provided the generalization gap bound of GCN, *i.e.*, $\mathcal{L}(f|\boldsymbol{X}, \boldsymbol{Y}, \boldsymbol{A}) - \hat{\mathcal{L}}(f|\boldsymbol{X}, \boldsymbol{Y}, \boldsymbol{A})$, by weights and biases parameter bounds. For simplicity, we restate the generalization gap bound of two-layer GCN without bias terms as in Remark 3.1. Here, we refer $\| \cdot \|_\infty$ be the maximum absolute sum of rows and $\| \cdot \|_{2 \to \infty}$ be the maximum 2-norm of columns.

*Remark* 3.1 (Generalization Gap Bound for GCN (Esser et al., 2021)). If $\boldsymbol{f} = \{f \subset \boldsymbol{f} | \|\boldsymbol{W}_k\|_\infty \leq \omega$ for $k \in \{1, 2\}\}$ and $\sigma$ is $L_\sigma$-Lipschitz continuous, the generalization gap bound for two-layer GCN $f$ satisfies

$$\mathcal{L}(f|\boldsymbol{X}, \boldsymbol{Y}, \boldsymbol{A}) - \hat{\mathcal{L}}(f|\boldsymbol{X}, \boldsymbol{Y}, \boldsymbol{A}) \leq \frac{nab}{m(n-m)} \|\bar{\boldsymbol{A}}\|_\infty \|\bar{\boldsymbol{A}}\boldsymbol{X}\|_{2 \to \infty} \sqrt{\log(n)} + O(n, m, \delta),$$

with probability $1 - \delta$ given certain small $\delta \in (0, 1)$, $a = 2L_\sigma\omega$, $b = \omega\sqrt{2/d_1}$, and $O(n, m, \delta)$ is certain function of $n, m, \delta$.

**DropEdge**  DropEdge (Rong et al., 2020) prevents over-fitting by randomly dropping edges during the training. This edge perturbation produces a new graph similar to the original graph so that it allows the model to be learned through diverse graphs. The only difference between the standard GNN learning procedure and DropEdge is the adjacency matrix. Let $\boldsymbol{A}$ be the original adjacency matrix, and $\boldsymbol{M}_p$ be the masking matrix that makes dropping edges with probability $p$. During the training, DropEdge replaces the adjacency matrix $\boldsymbol{A}$ with

$$\boldsymbol{A}_{\text{DropEdge}} = \boldsymbol{M}_p \circ \boldsymbol{A},$$

where $\circ$ is the Hadamard product of the matrix. Remark that, $\boldsymbol{M}_p$ is shared with all layers, and we apply the adjacency diffusion operator on $\boldsymbol{A}_{\text{DropEdge}}$.

## 4   OUR METHOD: IGRAPHMIX

In this section, we introduce iGraphMix, a simple yet effective Input Mixup method for node classification, as illustrated in Figure 1. To provide an intuitive understanding of the proposed method, we present a motivating example that involves two papers from various domains: one in machine learning and the other in chemistry, each with its own set of citation papers. When writing a new multidisciplinary paper (*i.e.*, mixing labels) that combines the two aforementioned papers, such as synthesizing chemical molecules via machine learning, a straightforward approach is to mix the main text (*i.e.*, mixing features) and cited papers (*i.e.*, mixing adjacency matrix) from both papers. This

simplistic example shows what iGraphMix aims to do. In Definition 4.1, we present the process of iGraphMix on node classification in a batch-wise manner.

**Definition 4.1** (iGraphMix for node classification). Let $M_\lambda$ be the masking matrix with $\lambda$ dropping probability. iGraphMix mixes feature matrix, one-hot label matrix, and adjacency matrix as follows:

$$
\begin{aligned}
\tilde{X} &:= \lambda X + (1 - \lambda) X', \\
\tilde{Y} &:= \lambda Y + (1 - \lambda) Y', \\
\tilde{A} &:= M_{1-\lambda} \circ A + M_\lambda \circ A',
\end{aligned}
\tag{4}
$$

where $(X', Y', A')$ is the permuted batch within labeled nodes of $(X, Y, A)$.

The most important point of mixing two nodes is blending the edges shown in the last equation of Eq. (4). The last equation in Eq. (4) may be quite different from our intuition. This is because we regard the masking matrix as a selecting neighbor matrix. In other words, $M_{1-\lambda}$ and $M_\lambda$ indicate that neighbors are chosen with probability $\lambda$ and $1 - \lambda$, respectively. Thus, in order to apply the importance $\lambda$ to neighbors, we have to multiply $M_{1-\lambda}$ by $A$ and vice versa. Here, the mixing coefficient $\lambda$ is drawn from $\mathrm{Beta}(\alpha, \alpha)$ as in Mixup (Zhang et al., 2018). This distribution is symmetric and becomes a uniform distribution when $\alpha = 1$. When $\alpha$ becomes smaller and smaller, $\lambda$ is sampled with values in the vicinity of zero or one. It means that the proposed method generates a new virtual node that is very similar to one of the original nodes when $\alpha$ is small. On the contrary, when $\alpha$ becomes larger and larger, $\lambda$ is sampled with values in the vicinity of 0.5. In this case, the proposed method produces a new virtual node in which two original nodes are evenly mixed. Note that $M_\lambda$ and $M_{1-\lambda}$ are shared with all layers. For the detailed implementation of iGraphMix in a Pytorch-like code, please refer to Appendix A.

## 5 THEORETICAL ANALYSIS

In this section, we theoretically analyze why GCNs trained with iGraphMix prevent over-fitting and generalize well compared to GCNs trained without data augmentation on the transductive node classification. Our theoretical analysis is inspired by Zhang et al. (2021) who analyzed Mixup in the generalization view, and Esser et al. (2021) who attempted to theoretically explain GCNs' behavior on the transductive node classification.

For the theoretical analysis, we consider a simple two-layer GCN with a point-wise ReLU activation function and one-dimensional output that classifies two classes -1 or 1. The GCN output logits of $m$ training nodes are

$$
\begin{aligned}
Z_{:m} &= \left( \bar{A} \sigma \left( \bar{A} X W_1 \right) W_2 \right)_{:m} \\
&= \left( \bar{A} H_1 W_2 \right)_{:m},
\end{aligned}
\tag{5}
$$

where $Z_{:m}$ is first $m$ rows of $Z$. We denote that $\tilde{Z}$ is from substituting node features and adjacency diffusion operator on Eq. (5) to those defined in Eq. (4). Also, $\tilde{Z}_{v,v'}$ states the $v$-th output logit interpolated $v$-th and $v'$-th nodes by iGraphMix. Then, we introduce the empirical loss for iGraphMix to train the GCN model in Eq. (5) as Definition 5.1.

**Definition 5.1** (Empirical loss on iGraphMix). Let $\lambda \sim \mathrm{Beta}(\alpha, \alpha)$, $M$ be the masking matrix. Then, the empirical loss for iGraphMix can be formulated as

$$
\hat{\mathcal{L}}(f | \tilde{X}, \tilde{Y}, \tilde{A}) = \frac{1}{m^2} \mathbb{E}_{\lambda, M} \left[ \sum_{v,v'=1}^{m} \ell \left( \tilde{Z}_{v,v'}, \tilde{Y}_{v,v'} \right) \right].
\tag{6}
$$

Note that the labeled nodes in Eq. (6) are the same as those in Eq. (3). It means that the only difference between them is using the Mixup dataset or not. Also, we consider the mean-square error (MSE) loss function for $v$-th node defined as $\ell(Z_v, Y_v) = \frac{1}{2} \| Z_v - Y_v \|^2$ (Zhang et al., 2021).

For the theoretical analysis, we suppose that there is no connection between labeled nodes. This assumption is quite reasonable for two-layer and the semi-supervised setting, *e.g.*, Citeseer dataset contains only 1.71% connected edges of labeled nodes out of all edges. Then, Lemma 5.2 shows that iGraphMix induces additional regularization of the trainable weights for the standard training.

**Lemma 5.2.** *Let* $\mathcal{R}(\boldsymbol{W}_2)$ *be the certain function of the second-order of* $\boldsymbol{W}_2$. *Then,* $\hat{\mathcal{L}}(f|\tilde{\boldsymbol{X}}, \tilde{\boldsymbol{Y}}, \tilde{\boldsymbol{A}}) \approx \hat{\mathcal{L}}(f|\boldsymbol{X}, \boldsymbol{Y}, \boldsymbol{A}) + \mathcal{R}(\boldsymbol{W}_2)$.

From the regularization point of view, we can reduce the weight space for iGraphMix with a related dual form of regularization term as

$$\boldsymbol{f}_{\text{iGraphMix}} := \{f \subset \boldsymbol{f} \mid \|\boldsymbol{W}_1\|_\infty \leq \omega, \text{and } \mathcal{R}(\boldsymbol{W}_2) \leq \omega\}, \tag{7}$$

where $\omega$ is the certain scalar values such that $\omega > 0$. Then, we provide the generalization gap bound with given weight space $\boldsymbol{f}_{\text{iGraphMix}}$ in Theorem 5.3.

**Theorem 5.3** (Generalization Gap Bound for GNN trained with iGraphMix). *For any* $f$ *in the weight space of iGraphMix* $\boldsymbol{f}_{\text{iGraphMix}}$, *we have the generalization gap bound as follows:*

$$\mathcal{L}(f|\boldsymbol{X}, \boldsymbol{Y}, \boldsymbol{A}) - \hat{\mathcal{L}}(f|\boldsymbol{X}, \boldsymbol{Y}, \boldsymbol{A}) \leq \frac{nabc}{m(n-m)}\|\bar{\boldsymbol{A}}\|_\infty\|\bar{\boldsymbol{A}}\boldsymbol{X}\|_{2\to\infty}\sqrt{\log(n)} + O(n, m, \delta),$$

*where* $a = 2L_\sigma\omega$, $b = \omega\sqrt{2/d_1}$, *and* $c = Q(\alpha, \boldsymbol{A}, \boldsymbol{X})$ *when* $Q(\cdot)$ *is the certain function of* $(\cdot)$.

Theorem 5.3 provides the evidence that weight space induced by iGraphMix could provide a tighter upper bound of generalization gap than standard training for the certain condition of beta distribution's parameter $\alpha$ and the data statistics $(\boldsymbol{A}, \boldsymbol{X})$. It implies that the optimal $\alpha$ could vary with respect to the graph characteristics. Thus, we easily confirm Corollary 5.4 that an upper bound of the generalization gap tends to be smaller in iGraphMix compared to standard training with appropriate $\alpha$.

**Corollary 5.4.** *Let* $f^*_{\text{std}}$ *and* $f^*_{\text{iGraphMix}}$ *be the optimally trained model by standard training and iGraphMix training respectively. Further,* $U(\cdot)$ *means the upper bound of* $(\cdot)$. *With appropriate* $\alpha$ *for* $\boldsymbol{A}$ *and* $\boldsymbol{X}$, *the following inequality holds with high probability.*

$$U\left(\mathcal{L}(f^*_{\text{std}}|\boldsymbol{X}, \boldsymbol{Y}, \boldsymbol{A}) - \hat{\mathcal{L}}(f^*_{\text{std}}|\boldsymbol{X}, \boldsymbol{Y}, \boldsymbol{A})\right) \geq U\left(\mathcal{L}(f^*_{\text{iGraphMix}}|\boldsymbol{X}, \boldsymbol{Y}, \boldsymbol{A}) - \hat{\mathcal{L}}(f^*_{\text{iGraphMix}}|\boldsymbol{X}, \boldsymbol{Y}, \boldsymbol{A})\right).$$

The detailed proofs are referred to Appendix B.

# 6 EXPERIMENTS

We compared the iGraphMix with five graph data augmentation methods: (1) *None* that trains GNNs with the graph which is not applied any augmentation methods; (2) *DropEdge* (Rong et al., 2020) that trains GNNs with the graph whose edges are randomly removed at each training epoch; (3) *DropNode* (Feng et al., 2020) that trains GNNs with the graph whose nodes are randomly masked at each training epoch; (4) *DropMessage* (Fang et al., 2023) that trains GNNs with perturbing propagated messages at each training epoch; (5) *M-Mixup* (Wang et al., 2021) that trains GNNs by interpolating nodes' hidden representations and corresponding labels. As some previous methods (Verma et al., 2021; Zhao et al., 2021; Liu et al., 2022a) require additional modifications to the model, auxiliary loss, and training techniques, there is potential for deviating from the authors' original intent while forcing them to other GNN models. Therefore, we excluded these methods from our experiments.

For the transductive node classification, We considered five datasets: CiteSeer, CORA, PubMed (Sen et al., 2008), ogbn-arxiv (Hu et al., 2020), and Flickr (McAuley & Leskovec, 2012). In CiteSeer, CORA, PubMed, and ogbn-arxiv, the nodes are the papers, and there are edges when one paper cites another paper. The goal of these datasets is to predict the subject class of each paper. Flickr is a dataset from an image-sharing SNS. In Flickr, nodes are images, and edges are connected when images share certain information, *e.g.*, common hashtag. It aims to predict the category or community class of nodes. Furthermore, conducted experiments on inductive node classification and link prediction tasks, and the results are shown in Appendix E. In short, our method experimentally outperformed comparative methods even for these tasks.

In order to demonstrate the superiority of various GNN models trained using iGraphMix over the baselines, this study selected three GNNs as the backbone models for evaluation: GCN (Kipf & Welling, 2017), GATv1 (Veličković et al., 2018), and GATv2 (Brody et al., 2022). We evaluated the generalization gap and the performance of iGraphMix and other augmentation baselines. Then, we examined the proposed method and other baselines by manipulating the number of layers and the number of nodes per class to show consistent improvement of our method. Lastly, we confirmed the versatility of iGraphMix by combining it with other augmentation methods. We refer to Appendix C for more details of experimental settings.

Table 1: Overall Micro-F1 score (%) on the datasets. The results are the average scores and standard deviations of ten trials with different random seeds. OOM means the out-of-memory.

| Backbone | Data Augmentation | Datasets | | | | |
|---|---|---|---|---|---|---|
| | | CiteSeer | CORA | PubMed | ogbn-arxiv | Flickr |
| GCN | None | 72.05 (0.56) | 82.65 (0.55) | 79.32 (0.15) | 67.11 (0.75) | 52.77 (0.14) |
| | DropEdge | 72.07 (0.28) | 83.20 (0.07) | 79.38 (0.19) | 68.17 (0.39) | 53.59 (0.07) |
| | DropNode | 72.48 (0.44) | 82.65 (0.23) | 79.52 (0.11) | 67.61 (1.17) | 53.29 (0.18) |
| | DropMessage | 73.35 (0.46) | 83.40 (0.61) | 79.60 (0.32) | 68.71 (0.30) | 53.55 (0.10) |
| | M-Mixup | 71.52 (0.80) | 80.28 (0.66) | 78.63 (0.32) | 65.82 (0.61) | 48.14 (0.25) |
| | iGraphMix (*ours*) | **73.67 (0.61)** | **83.78 (0.42)** | **79.93 (0.60)** | **68.93 (0.35)** | **53.61 (0.12)** |
| GATv1 | None | 71.14 (1.13) | 79.98 (0.69) | 77.75 (0.63) | 65.08 (1.03) | 52.07 (0.28) |
| | DropEdge | 71.45 (1.02) | 82.65 (0.60) | 77.85 (0.28) | 67.98 (0.40) | 53.07 (0.21) |
| | DropNode | 70.78 (0.65) | 81.33 (1.00) | 77.11 (0.75) | 67.49 (0.65) | 53.05 (0.25) |
| | DropMessage | 72.20 (0.52) | **83.39 (0.60)** | 78.00 (0.55) | 68.45 (0.35) | 52.95 (0.30) |
| | M-Mixup | 72.02 (0.64) | 82.06 (0.85) | **78.93 (0.58)** | OOM | 52.09 (0.20) |
| | iGraphMix (*ours*) | **72.28 (0.60)** | 83.20 (0.63) | 78.41 (0.31) | **69.49 (0.41)** | **53.22 (0.18)** |
| GATv2 | None | 70.41 (1.91) | 79.11 (0.80) | 77.87 (0.51) | 65.63 (0.75) | 51.79 (0.43) |
| | DropEdge | 71.15 (0.78) | 82.20 (0.62) | 77.94 (0.51) | 67.84 (0.74) | 53.23 (0.21) |
| | DropNode | 70.33 (1.59) | 80.44 (1.84) | 77.90 (0.42) | 68.64 (0.61) | 53.38 (0.61) |
| | DropMessage | 71.49 (1.02) | 82.27 (1.28) | 78.19 (0.63) | 68.66 (0.45) | 53.06 (0.23) |
| | M-Mixup | **72.90 (1.47)** | 82.06 (0.91) | 78.36(0.55) | OOM | 52.38 (0.20) |
| | iGraphMix (*ours*) | 71.97 (0.66) | **82.80 (0.49)** | **78.73 (0.33)** | **69.82 (0.31)** | **53.90 (0.20)** |

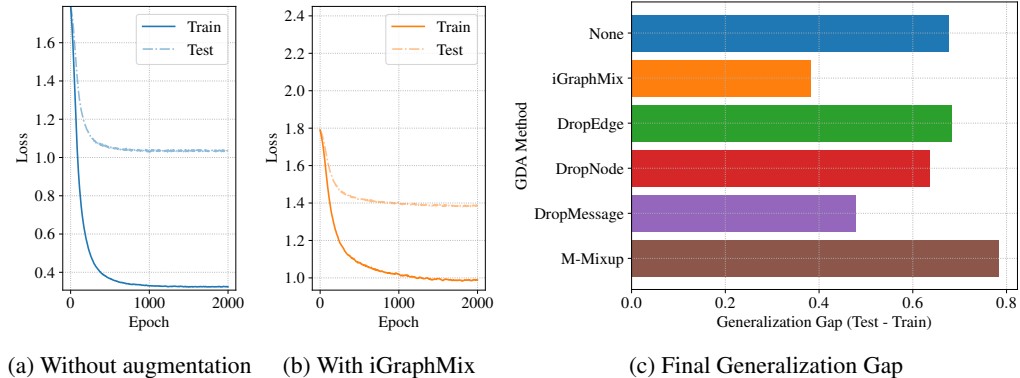

(a) Without augmentation  (b) With iGraphMix  (c) Final Generalization Gap

Figure 2: Generalization ability of GCN. (a)-(b): Train and test loss with respect to the training epochs ((a): Without augmentation, (b): With iGraphMix). (c): Final generalization gap over all augmentation methods.

## 6.1 OVERALL RESULTS

The overall results on node classification are summarized in Table 1. We compared iGraphMix with five graph data augmentation methods in terms of the test micro-F1 score of models at the best validation micro-F1 score epoch. We found that the proposed method outperforms the models without data augmentation by 2.84% on average. Reminding Lemma 5.2, this result may indicate that iGraphMix is the effective regularization method to improve the test micro-F1 score in the various datasets and backbone models. We also confirmed that iGraphMix outperforms DropEdge, DropNode, and DropMessage across the five datasets and three backbone models. This result may imply that iGraphMix generates augmented graph data that mitigates the distorted interrelationship between augmented graphs and labels better than DropEdge, DropNode, and DropMessage do. In addition, from the result that iGraphMix outperformed M-Mixup by an average of 2.41%, we may infer that iGraphMix generates more diverse virtual nodes from the vicinity of original nodes than M-Mixup, preventing the over-fitting problem more effectively.

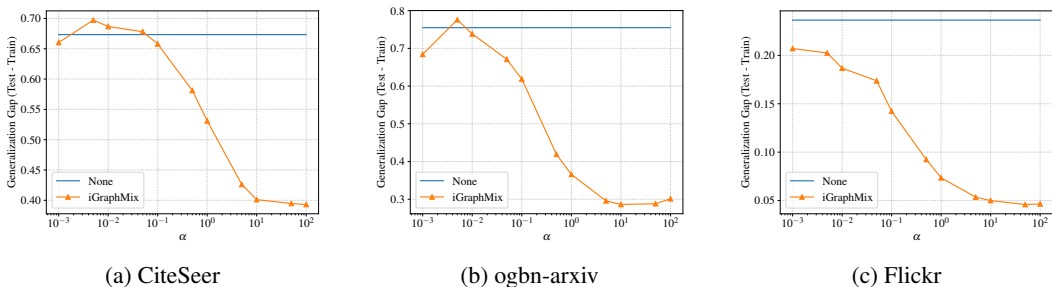

(a) CiteSeer                    (b) ogbn-arxiv                    (c) Flickr

Figure 3: Generalization gap with respect to Beta distribution's parameter $\alpha$ on various three datasets.

## 6.2 GENERALIZATION GAP

**Compared with Baselines**    We examined whether iGraphMix can improve the generalization ability by reducing the generalization gap analyzed in Section 5. Figure 2 shows the generalization gap of GCNs trained with six augmentation methods on CiteSeer. Compared the loss in Figures 2a and 2b, we found that the gap between the test and the training loss of GCN trained with iGraphMix is consistently smaller than that of GCN trained without augmentation methods. This result empirically supports our theoretical finding that iGraphMix improves the generalization ability of GNN. We compared the generalization gap of iGraphMix with other augmentation methods in Figure 2c. It empirically shows that GNNs trained with iGraphMix have a smaller or comparable generalization gap than GNNs trained with other augmentation methods. We also found the similar results in other datasets as shown in Appendix D.1.

**Beta Distribution Parameters $\alpha$**    We verified Corollary 5.4 that the generalization gap becomes smaller depending on the appropriate beta distribution parameter $\alpha$ for the graph characteristics. Beta distribution parameter $\alpha$ controls the amount of diverse augmented graphs of iGraphMix for training. When we set $\alpha$ small, iGraphMix is likely to generate graphs similar to the original graph. Conversely, when $\alpha$ is large, iGraphMix is likely to generate well-mixed graphs. The generalization gap with respect to the parameter $\alpha$ is illustrated in Figure 3. We confirmed that the performance of iGraphMix, which utilizes the data augmentation approach, is similar to the method without augmentation when $\alpha$ is near zero. Also, we found that the generalization gap of iGraphMix becomes smaller when $\alpha$ is large. This result reveals that the generalization ability becomes better when well-mixed graphs are used in training. In addition, by comparing the optimal generalization gap in CiteSeer, ogbn-arxiv, and Flickr in Figure 3, we verified that the optimal $\alpha$ of these datasets are 100, 10, and 50, respectively. This result empirically supports Corollary 5.4 that the different datasets require different optimal $\alpha$ to achieve the small generalization gap.

## 6.3 ANALYSIS ON VARIOUS SETTINGS

In this section, we examined how manipulating the number of layers and labeled nodes affects the performance of iGraphMix. The results of GCN on the CiteSeer are presented and analyzed. Comprehensive results obtained from other datasets and models can be found in Appendix D.2.

**Number of Layers $K$**    We evaluated the micro-F1 score depending on the number of layers $K$. Increasing $K$ of GNN leads to the over-smoothing problem (Li et al., 2018), as node embeddings become indistinguishable by considering the larger-hop relationship between nodes. Thus, this experiment also tested that the proposed method mitigates the over-smoothing problem. Figure 4a shows the performance of the proposed method for the different $K$. We found that training GCN with iGraphMix improves the micro-F1 score by 2.33%, 6.25%, and 6.39% for two-, four-, and eight-layer GCNs respectively compared to that without augmentations. This result may imply that our method improves the micro-F1 score of GNNs with various $K$. More importantly, the results indicate that training GNNs with our method may diminish the over-smoothing problem regardless of $K$. Since iGraphMix changes connected edges and corresponding labels in each iteration, the proposed method can mitigate the representation collapse of connected nodes at each training step.

**Number of Labeled Nodes per Class $L$**    As illustrated in Figure 4b, we assessed the micro-F1 score of the proposed method by changing the number of labeled nodes per class $L$. We confirmed

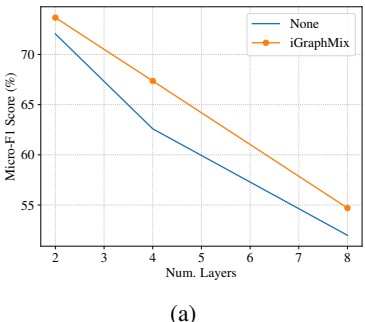 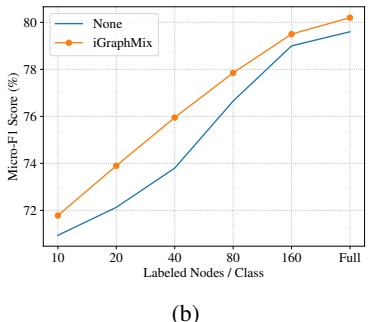

(a)  (b)

Figure 4: Analysis of iGraphMix on the various number of (a) layers and (b) nodes per class by Micro-F1 score (%).

that training GCN with iGraphMix increases the micro-F1 score compared to that without augmentations by 1.41%, 2.33%, 5.22%, 1.39%, 0.80%, and 0.61% for $L = 10, 20, 40, 80, 160,$ and Full, respectively. Notably, we found that our method can be more effective in improving the micro-F1 score when training with fewer $L$ than fully labeled nodes. This result may imply that iGraphMix alleviates the over-fitting problem, especially for the lack of labeled data.

## 6.4 COMBINATION WITH OTHER AUGMENTATION METHODS

We verified the versatility of our method by applying iGraphMix after other augmentation methods were employed at each training step. Table 2 shows the comparison between the performance of None, DropEdge, DropNode, and DropMixup with that of unifying iGraphMix and them on CiteSeer with 2-layer GCN. We found that unifying iGraphMix with other augmentation methods leads to performance improvement with 1.86% on average. This result shows that iGraphMix is versatile with other augmentation methods to boost the performance of GNNs, similar to Berthelot et al. (2019).

Table 2: Micro-F1 score (%) of unifying iGraphMix with other augmentations on CiteSeer.

| Data Augmentation Methods | | | | Performance |
|---|---|---|---|---|
| DropEdge | DropNode | DropMessage | iGraphMix | |
| | | | | 72.05 |
| | | | ✓ | **73.67** |
| ✓ | | | | 72.07 |
| ✓ | | | ✓ | **73.97** |
| | ✓ | | | 72.48 |
| | ✓ | | ✓ | **73.75** |
| | | ✓ | | 73.35 |
| | | ✓ | ✓ | **73.93** |

## 7 CONCLUSION

**Summary**  This paper proposed iGraphMix that addresses the irregularity and alignment issues of Input Mixup on node classification. Specifically, to address the two issues, iGraphMix does not only interpolate node features and labels but also aggregates the sampled neighboring nodes. Theoretical analysis of the generalization gap and our experiments on the real-world graphs showed that the proposed method is effective in regularizing GNNs by generating diverse virtual samples and preserving high usability and versatility.

**Future Works**  There are two possible directions for future works. The first direction is to find a better edge sampling method for Input Mixup on node classification, similar to the research on Input Mixup in the other domains (Kim et al., 2020; Kong et al., 2022). The second direction is to combine improvements of other techniques in graph learning, such as pseudo-labeling (Verma et al., 2021) and consistency loss (Feng et al., 2020), with iGraphMix. We hope this work could be the crucial step to improving Input Mixup on node classification.

## ETHICS STATEMENT

Our method may not consider mitigating biases or inequalities. Our work initially focused on the technical aspects and did not explicitly address potential biases or inequalities that might arise. However, we believe that there can be a potential future direction to address biases and inequalities in algorithms based on our method.

## REPRODUCIBILITY STATEMENT

Our method is built on Pytorch 1.12.1. (Paszke et al., 2019) and Pytorch Geometric 2.1.0 (Fey & Lenssen, 2019). The licenses of Pytorch and Pytorch Geometric are available under BSD-style and MIT respectively. Our experiments were conducted on NVIDIA V100 with CUDA version 11.3. Refer to the appendices for further reproducibility details, such as code, hyper-parameters, and so on.

## ACKNOWLEDGEMENT

We appreciate Youngin Cho from NCSOFT Co. and Moonseok Choi from KAIST for their valuable feedback on this paper.

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

# APPENDICES

In the appendix, we provide the implementation details of iGraphMix, precise proof of our theoretical findings, and additional experiments on various settings that are not shown in the main text due to space limitations.

# A    IMPLEMENTATION DETAILS OF IGRAPHMIX

We provide the PyTorch-like style implementation of iGraphMix for node classification in Algorithm 1.

---

**Algorithm 1** iGraphMix: Pytorch-like Implementation with Torch Geometric.

```python
import numpy as np
import torch
from torch_geometric.data import Data
from torch_geometric.utils import dropout_adj

# data: Torch-geometric Graph Data instance consists of node features, edges, and labels.
# alpha: Beta distribution hyper-parameter
# batch_size: Number of train nodes

def i_graph_mix(data, alpha, batch_size):
    # 0) Sample lambda
    lambda_ = np.random.beta(alpha, alpha)

    # 1) Index Permutation
    index = torch.randperm(batch_size)
    index = torch.cat((index, torch.arange(batch_size, data.x.shape[0])), dim=0)
    index_new = torch.zeros(index.shape[0], dtype=torch.long)
    index_new[index] = torch.arange(0, index.shape[0])

    # 2) Node feature Mixup
    x = data.x.clone().detach()
    x_mixup = lambda_ * x + (1-lambda_) * x[index]

    # 3) Label Mixup
    y = data.y.clone().detach()
    y_mixup = lambda_ * y + (1-lambda_) * y[index]

    # 4) Edge Mixup
    edge_index = data.edge_index.clone().detach()
    row, col = edge_index.clone().detach()[0], edge_index.clone().detach()[1]
    row, col = index_new.clone().detach()[row], index_new.clone().detach()[col]
    edge_index_perm = torch.stack([row, col], dim=0)

    # 5) Prepare Mixup Graph
    edge_index, _ = dropout_adj(edge_index, p=1-lambda_, training=True)
    edge_index_perm, _ = dropout_adj(edge_index_perm, p=lambda_, training=True)
    edge_index_mixup = torch.cat((edge_index, edge_index_perm), dim=1)
    data_mixup = Data(x=x_mixup, y=y_mixup, edge_index=edge_index_mixup)
    data_mixup.train_mask = data.train_mask
    data_mixup.val_mask = data.val_mask
    data_mixup.test_mask = data.test_mask

    return data_mixup

def train(model, data, loss_func, optimizer, alpha, batch_size):
    model.train()
    # 1) Mix-up graph
    data_mixup = i_graph_mix(data, alpha, batch_size)

    # 2) Model Inference
    output = model(data_mixup.x, data_mixup.edge_index)[:batch_size]

    # 3) Calculate and optimize loss
    loss = loss_func(output, data_mixup.y[:batch_size])
    loss.backward()
    optimizer.step()
```

---

## B  PROOF IN THEORETICAL ANALYSIS

This section provides the proof details of theoretical findings. We state $\boldsymbol{A}_v$ and $\boldsymbol{A}_{:,v}$ be $v$-th row and column vector of the matrix $\boldsymbol{A}$ respectively. For the theoretical analysis, we consider the adjacency diffusion operator $\bar{\boldsymbol{A}} := \boldsymbol{A} + \boldsymbol{I}$.

As in the Section 5, we assume that there is no connection between labeled train nodes. It means that $\boldsymbol{A}\boldsymbol{X} = \boldsymbol{A}\boldsymbol{X}' = \boldsymbol{A}\tilde{\boldsymbol{X}}$ as mixup only applies at labeled nodes. Thus, this assumption can induce the following equation.

$$
\begin{aligned}
(\tilde{\boldsymbol{A}} + \boldsymbol{I})_v \tilde{\boldsymbol{X}} &= (\boldsymbol{M}_{1-\lambda} \circ \boldsymbol{A})_v \tilde{\boldsymbol{X}} + \boldsymbol{M}_\lambda \circ \boldsymbol{A}'_v \tilde{\boldsymbol{X}} + \boldsymbol{I}_v \tilde{\boldsymbol{X}} \\
&= (\boldsymbol{M}_{1-\lambda} \circ \boldsymbol{A})_v \boldsymbol{X} + \boldsymbol{M}_\lambda \circ \boldsymbol{A}'_v \boldsymbol{X}' + \tilde{\boldsymbol{X}}_v.
\end{aligned} \tag{8}
$$

Then, we derive Lemma B.1 to change the output logits in Eq. (5) from the general form to the iGraphMix form.

**Lemma B.1.** *Given any random variable $\lambda$, the permutation $\boldsymbol{P}$, and the masking matrix $\boldsymbol{M}$, we have*

$$
\mathbb{E}_{\lambda, \boldsymbol{P}, \boldsymbol{M}} \left[ \tilde{\boldsymbol{Z}}_{:m} \right] = \mathbb{E}_{\lambda, \boldsymbol{P}, \boldsymbol{M}} \left[ \left( (\tilde{\boldsymbol{A}} + \boldsymbol{I}) \tilde{\boldsymbol{H}}_1 \boldsymbol{W}_2 \right)_{:m} \right],
$$

*when there is no connection between labeled train nodes, and $\tilde{\boldsymbol{H}}_1 := \lambda \boldsymbol{H}_1 + (1 - \lambda) \boldsymbol{H}'_1$.*

*Proof.* $\mathbb{E}_{\lambda, \boldsymbol{P}, \boldsymbol{M}} \left[ \tilde{\boldsymbol{Z}}_{:m} \right]$ can be formulated as follows by Eq. (8).

$$
\begin{aligned}
\mathbb{E}_{\lambda, \boldsymbol{P}, \boldsymbol{M}} \left[ \tilde{\boldsymbol{Z}}_{:m} \right] &= \mathbb{E}_{\lambda, \boldsymbol{P}, \boldsymbol{M}} \left[ \left( (\tilde{\boldsymbol{A}} + \boldsymbol{I}) \sigma \left( (\tilde{\boldsymbol{A}} + \boldsymbol{I}) \tilde{\boldsymbol{X}} \boldsymbol{W}_1 \right) \boldsymbol{W}_2 \right)_{:m} \right] \\
&= \mathbb{E}_{\lambda, \boldsymbol{P}, \boldsymbol{M}} \left[ \left( (\tilde{\boldsymbol{A}} + \boldsymbol{I}) \sigma \left( \left( \boldsymbol{M}_{1-\lambda} \circ \boldsymbol{A}\boldsymbol{X} + \boldsymbol{M}_\lambda \circ \boldsymbol{A}'\boldsymbol{X}' + \tilde{\boldsymbol{X}} \right) \boldsymbol{W}_1 \right) \boldsymbol{W}_2 \right)_{:m} \right] \\
&= \mathbb{E}_{\lambda, \boldsymbol{P}, \boldsymbol{M}} \left[ \left( (\tilde{\boldsymbol{A}} + \boldsymbol{I}) \sigma \left( (\boldsymbol{M}_{1-\lambda} \circ \boldsymbol{A}\boldsymbol{X} + \boldsymbol{M}_\lambda \circ \boldsymbol{A}'\boldsymbol{X}' + \lambda \boldsymbol{X} + (1 - \lambda) \boldsymbol{X}') \boldsymbol{W}_1 \right) \boldsymbol{W}_2 \right)_{:m} \right] \\
&= \mathbb{E}_{\lambda, \boldsymbol{P}, \boldsymbol{M}} \left[ \left( (\tilde{\boldsymbol{A}} + \boldsymbol{I}) \sigma \left( (((\boldsymbol{M}_{1-\lambda} \circ \boldsymbol{A} + \lambda \boldsymbol{I}) \boldsymbol{X} + (\boldsymbol{M}_\lambda \circ \boldsymbol{A}' + (1 - \lambda)\boldsymbol{I}) \boldsymbol{X}') \boldsymbol{W}_1 \right) \boldsymbol{W}_2 \right)_{:m} \right] \\
&= \mathbb{E}_{\lambda, \boldsymbol{P}, \boldsymbol{M}} \left[ \left( (\tilde{\boldsymbol{A}} + \boldsymbol{I}) \sigma \left( (((\lambda \boldsymbol{A} + \lambda \boldsymbol{I}) \boldsymbol{X} + ((1 - \lambda)\boldsymbol{A}' + (1 - \lambda)\boldsymbol{I}) \boldsymbol{X}') \boldsymbol{W}_1 \right) \boldsymbol{W}_2 \right)_{:m} \right] \\
&= \mathbb{E}_{\lambda, \boldsymbol{P}, \boldsymbol{M}} \left[ \left( (\tilde{\boldsymbol{A}} + \boldsymbol{I}) \sigma \left( \lambda \bar{\boldsymbol{A}} \boldsymbol{X} \boldsymbol{W}_1 + (1 - \lambda) \bar{\boldsymbol{A}}' \boldsymbol{X}' \boldsymbol{W}_1 \right) \boldsymbol{W}_2 \right)_{:m} \right] \\
&= \mathbb{E}_{\lambda, \boldsymbol{P}, \boldsymbol{M}} \left[ \left( (\tilde{\boldsymbol{A}} + \boldsymbol{I}) \left( \lambda \sigma \left( \bar{\boldsymbol{A}} \boldsymbol{X} \boldsymbol{W}_1 \right) + (1 - \lambda) \sigma \left( \bar{\boldsymbol{A}}' \boldsymbol{X}' \boldsymbol{W}_1 \right) \right) \boldsymbol{W}_2 \right)_{:m} \right] \tag{9} \\
&= \mathbb{E}_{\lambda, \boldsymbol{P}, \boldsymbol{M}} \left[ \left( (\tilde{\boldsymbol{A}} + \boldsymbol{I}) \left( \lambda \boldsymbol{H}_1 + (1 - \lambda) \boldsymbol{H}'_1 \right) \boldsymbol{W}_2 \right)_{:m} \right] \tag{10} \\
&= \mathbb{E}_{\lambda, \boldsymbol{P}, \boldsymbol{M}} \left[ \left( (\tilde{\boldsymbol{A}} + \boldsymbol{I}) \tilde{\boldsymbol{H}}_1 \boldsymbol{W}_2 \right)_{:m} \right], \tag{11}
\end{aligned}
$$

where Eq. (9) holds because $\sigma$ has a property that $\sigma(\lambda z) = \lambda \sigma(z)$ for any $z \in \mathbb{R}$, Eq. (10) is from definition of hidden feature, Eq. (11) is from the definition of mixed hidden feature. $\square$

Also, we introduce Lemma B.2 that the mean of the aggregated representation from iGraphMix is the same as the mean of the original graph's aggregated representation. It is utilized in the proof of Lemma 5.2 and Theorem 5.3.

**Lemma B.2.** *Consider any* $\lambda \sim P_\lambda$, $\boldsymbol{P}$ *be the random permutation, i.e.* $\boldsymbol{H}_1' = \boldsymbol{P}\boldsymbol{H}_1$ *and* $\boldsymbol{A}' = \boldsymbol{P}\bar{\boldsymbol{A}}\boldsymbol{P}^\top$, *and* $\boldsymbol{M}$ *be the masking matrix. Then,* $1/m \sum_{v=1}^{m} \mathbb{E}_{\boldsymbol{P},\boldsymbol{M}} \left[ (\tilde{\boldsymbol{A}} + \boldsymbol{I})_v \tilde{\boldsymbol{H}}_1 \right] = 1/m \sum_{v=1}^{m} \bar{\boldsymbol{A}}_v \boldsymbol{H}_1$.

*Proof.* We begin with reformulate $1/m \sum_{v=1}^{m} \bar{\boldsymbol{A}}_v \boldsymbol{H}_1$ with $\tilde{\boldsymbol{H}}_1$ and $(\tilde{\boldsymbol{A}} + \boldsymbol{I})$.

$$
\begin{aligned}
\frac{1}{m} \sum_{v=1}^{m} \left( \bar{\boldsymbol{A}}\boldsymbol{H}_1 \right)_v &= \frac{1}{m} \sum_{v=1}^{m} \left( \bar{\boldsymbol{A}}\boldsymbol{H}_1 \right)_v \\
&= \frac{1}{m} \sum_{v=1}^{m} \left[ \lambda \left( \bar{\boldsymbol{A}}\boldsymbol{H}_1 \right)_v + (1 - \lambda) \left( \bar{\boldsymbol{A}}\boldsymbol{H}_1 \right)_v \right] \\
&= \frac{1}{m} \sum_{v=1}^{m} \mathbb{E}_{\boldsymbol{P}} \left[ \lambda \left( \bar{\boldsymbol{A}}\boldsymbol{H}_1 \right)_v + (1 - \lambda) \left( \boldsymbol{P}\bar{\boldsymbol{A}}\boldsymbol{H}_1 \right)_v \right] \\
&= \frac{1}{m} \sum_{v=1}^{m} \mathbb{E}_{\boldsymbol{P}} \left[ \left( \lambda\bar{\boldsymbol{A}}\boldsymbol{H}_1 + (1 - \lambda)\boldsymbol{P}\bar{\boldsymbol{A}}\boldsymbol{P}^\top\boldsymbol{P}\boldsymbol{H}_1 \right)_v \right] && (12) \\
&= \frac{1}{m} \sum_{v=1}^{m} \mathbb{E}_{\boldsymbol{P}} \left[ \left( \lambda\bar{\boldsymbol{A}}\boldsymbol{H}_1 + (1 - \lambda)\bar{\boldsymbol{A}}'\boldsymbol{H}_1' \right)_v \right] && (13) \\
&= \frac{1}{m} \sum_{v=1}^{m} \mathbb{E}_{\boldsymbol{P}} \Big[ \Big( \lambda\bar{\boldsymbol{A}}(\lambda\boldsymbol{H}_1) + \lambda\bar{\boldsymbol{A}}((1 - \lambda)\boldsymbol{H}_1') \\
&\qquad + (1 - \lambda)\bar{\boldsymbol{A}}'\lambda\boldsymbol{H} + (1 - \lambda)\bar{\boldsymbol{A}}'(1 - \lambda)\boldsymbol{H}_1' \Big)_v \Big] \\
&\qquad + \frac{1}{m} \sum_{v=1}^{m} \mathbb{E}_{\boldsymbol{P}} \left[ \left( \lambda\bar{\boldsymbol{A}}((1 - \lambda)(\boldsymbol{H}_1 - \boldsymbol{H}_1')) + (1 - \lambda)\bar{\boldsymbol{A}}'\lambda(\boldsymbol{H}_1' - \boldsymbol{H}_1) \right)_v \right], && (14)
\end{aligned}
$$

where Eq. (12) is due to normal matrix property for permutation matrix, Eq. (13) is due to the definition of $\boldsymbol{A}'$, $\boldsymbol{H}'$ based on permutation matrix. Since $1/m \sum_{v=1}^{m} \left[ \mathbb{E}_{v'} \left[ (\boldsymbol{H}_1 - \boldsymbol{H}_1')_v \right] \right] = 0$, Eq. (14) becomes 0. Then,

$$
\begin{aligned}
\frac{1}{m} \sum_{v=1}^{m} \left( \bar{\boldsymbol{A}}\boldsymbol{H}_1 \right)_v &= \frac{1}{m} \sum_{v}^{m} \mathbb{E}_{\boldsymbol{P}} \left[ \left( \lambda\bar{\boldsymbol{A}} \left( \lambda\boldsymbol{H}_1 + (1 - \lambda)\boldsymbol{H}_1' \right) + (1 - \lambda)\bar{\boldsymbol{A}}' \left( \lambda\boldsymbol{H}_1 + (1 - \lambda)\boldsymbol{H}_1' \right) \right)_v \right] \\
&= \frac{1}{m} \sum_{v=1}^{m} \mathbb{E}_{\boldsymbol{P}} \left[ \left( \lambda\bar{\boldsymbol{A}}\tilde{\boldsymbol{H}}_1 + (1 - \lambda)\bar{\boldsymbol{A}}'\tilde{\boldsymbol{H}}_1 \right)_v \right] \\
&= \frac{1}{m} \sum_{v=1}^{m} \mathbb{E}_{\boldsymbol{P}} \left[ \left( (\lambda\bar{\boldsymbol{A}} + (1 - \lambda)\bar{\boldsymbol{A}}')\tilde{\boldsymbol{H}}_1 \right)_v \right] \\
&= \frac{1}{m} \sum_{v=1}^{m} \mathbb{E}_{\boldsymbol{P}} \left[ \left( (\lambda\boldsymbol{A} + (1 - \lambda)\boldsymbol{A}' + \boldsymbol{I})\tilde{\boldsymbol{H}}_1 \right)_v \right] \\
&= \frac{1}{m} \sum_{v=1}^{m} \mathbb{E}_{\boldsymbol{P},\boldsymbol{M}} \left[ \left( (\tilde{\boldsymbol{A}} + \boldsymbol{I})\tilde{\boldsymbol{H}}_1 \right)_v \right] && (15) \\
&= \frac{1}{m} \sum_{v=1}^{m} \mathbb{E}_{\boldsymbol{P},\boldsymbol{M}} \left[ (\tilde{\boldsymbol{A}} + \boldsymbol{I})_v \tilde{\boldsymbol{H}}_1 \right],
\end{aligned}
$$

where Eq. (15) is due to the following equations.

$$\lambda \boldsymbol{A} + (1-\lambda)\boldsymbol{A}' = \lambda \mathbf{1} \circ \boldsymbol{A} + (1-\lambda)\mathbf{1} \circ \boldsymbol{A}'$$
$$= \mathbb{E}_{\boldsymbol{M}}\left[\boldsymbol{M}_{1-\lambda} \circ \boldsymbol{A} + \boldsymbol{M}_{\lambda} \circ \boldsymbol{A}'\right]$$
$$:= \mathbb{E}_{\boldsymbol{M}}\left[\tilde{\boldsymbol{A}}\right].$$

$\square$

### B.1 PROOF OF LEMMA 5.2

Before providing the proof of Lemma 5.2, we show Lemma B.3 and Lemma B.4 to get the approximate form of iGraphMix loss.

**Lemma B.3.** *Let* $\tilde{D}_\lambda := \frac{\alpha}{\alpha+\beta}\mathrm{Beta}(\alpha+1,\beta) + \frac{\beta}{\alpha+\beta}\mathrm{Beta}(\beta+1,\alpha)$, $\boldsymbol{P}$ *be the permutation induced by the randomness for mixed target samples,* $\boldsymbol{M}$ *be the masking matrix. Also, we consider* $\check{\boldsymbol{Z}}_v$ *be the* $v$-th *virtual node output logits given* $\tilde{\boldsymbol{X}}$ *and* $\tilde{\boldsymbol{A}}$ *that are followed by the distribution* $\tilde{D}_\lambda$ *and the permutation* $\boldsymbol{P}$. *Then, the empirical loss is converted as follows.*

$$\hat{\mathcal{L}}(f|\tilde{\boldsymbol{X}},\tilde{\boldsymbol{Y}},\tilde{\boldsymbol{A}}) = \frac{1}{m}\sum_{v=1}^{m}\mathbb{E}_{\lambda'\sim\tilde{D}_\lambda,\boldsymbol{M}}\mathbb{E}_{\boldsymbol{P}}\left[\ell(\check{\boldsymbol{Z}}_v,\boldsymbol{Y}_v)\right].$$

*Proof.*

$$\hat{\mathcal{L}}(f|\tilde{\boldsymbol{X}},\tilde{\boldsymbol{Y}},\tilde{\boldsymbol{A}}) = \frac{1}{m^2}\sum_{v,v'=1}^{m}\mathbb{E}_{\lambda,\boldsymbol{M}}\left[\ell\left(\tilde{\boldsymbol{Z}}_{v,v'},\tilde{\boldsymbol{Y}}_{v,v'}\right)\right]$$

$$= \frac{1}{m^2}\sum_{v,v'=1}^{m}\mathbb{E}_{\lambda,\boldsymbol{M}}\left[\|\tilde{\boldsymbol{Z}}_{v,v'} - \tilde{\boldsymbol{Y}}_{v,v'}\|^2\right]$$

$$= \frac{1}{m^2}\sum_{v,v'=1}^{m}\mathbb{E}_{\lambda,\boldsymbol{M}}\left[\tilde{\boldsymbol{Z}}_{v,v'}^\top\tilde{\boldsymbol{Z}}_{v,v'} - 2\tilde{\boldsymbol{Y}}_{v,v'}^\top\tilde{\boldsymbol{Z}}_{v,v'} + \tilde{\boldsymbol{Y}}_{v,v'}^\top\tilde{\boldsymbol{Y}}_{v,v'}\right]$$

$$= \frac{1}{m^2}\sum_{v,v'=1}^{m}\mathbb{E}_{\lambda,\boldsymbol{M}}\left[\tilde{\boldsymbol{Z}}_{v,v'}^\top\tilde{\boldsymbol{Z}}_{v,v'} - 2\left(\lambda\boldsymbol{Y}_v + (1-\lambda)\boldsymbol{Y}_{v'}\right)^\top\tilde{\boldsymbol{Z}}_{v,v'} + \tilde{\boldsymbol{Y}}_{v,v'}^\top\tilde{\boldsymbol{Y}}_{v,v'}\right]$$

$$= \frac{1}{m^2}\sum_{v,v'=1}^{m}\mathbb{E}_{\lambda,\boldsymbol{M}}\left[\lambda\left(\tilde{\boldsymbol{Z}}_{v,v'}^\top\tilde{\boldsymbol{Z}}_{v,v'} - 2\boldsymbol{Y}_v^\top\tilde{\boldsymbol{Z}}_{v,v'} + \boldsymbol{Y}_v^\top\boldsymbol{Y}_v\right)\right.$$

$$\left. + (1-\lambda)\left(\tilde{\boldsymbol{Z}}_{v,v'}^\top\tilde{\boldsymbol{Z}}_{v,v'} - 2\boldsymbol{Y}_{v'}^\top\tilde{\boldsymbol{Z}}_{v,v'} + \boldsymbol{Y}_{v'}^\top\boldsymbol{Y}_{v'}\right)\right]$$

$$+ \frac{1}{m^2}\sum_{v,v'=1}^{m}\mathbb{E}_{\lambda,\boldsymbol{M}}\left[\lambda\boldsymbol{Y}_v^\top(\tilde{\boldsymbol{Y}}_{v,v'} - \boldsymbol{Y}_v) + (1-\lambda)\boldsymbol{Y}_{v'}^\top(\tilde{\boldsymbol{Y}}_{v,v'} - \boldsymbol{Y}_{v'})\right]$$

$$= \frac{1}{m^2}\sum_{v,v'=1}^{m}\mathbb{E}_{\lambda,\boldsymbol{M}}\left[\mathbb{E}_{Q\sim\mathrm{Bern}(\lambda)}\left[Q\left(\tilde{\boldsymbol{Z}}_{v,v'}^\top\tilde{\boldsymbol{Z}}_{v,v'} - 2\boldsymbol{Y}_v^\top\tilde{\boldsymbol{Z}}_{v,v'} + \boldsymbol{Y}_v^\top\boldsymbol{Y}_v\right)\right]\right]$$

$$+ \frac{1}{m^2}\sum_{v,v'=1}^{m}\mathbb{E}_{\lambda,\boldsymbol{M}}\left[\mathbb{E}_{Q\sim\mathrm{Bern}(\lambda)}\left[(1-Q)\left(\tilde{\boldsymbol{Z}}_{v,v'}^\top\tilde{\boldsymbol{Z}}_{v,v'} - 2\boldsymbol{Y}_{v'}^\top\tilde{\boldsymbol{Z}}_{v,v'} + \boldsymbol{Y}_{v'}^\top\boldsymbol{Y}_{v'}\right)\right]\right]$$

(16)

$$= \frac{1}{m^2}\sum_{v,v'=1}^{m}\mathbb{E}_{Q\sim\mathrm{Bern}(\frac{\alpha}{\alpha+\beta}),\boldsymbol{M}}\left[\mathbb{E}_{\lambda\sim\mathrm{Beta}(\alpha+Q,\beta+1-Q)}\left[Q\ell(\tilde{\boldsymbol{Z}}_{v,v'},\boldsymbol{Y}_v)\right]\right]$$

$$+ \frac{1}{m^2}\sum_{v,v'=1}^{m}\mathbb{E}_{Q\sim\mathrm{Bern}(\frac{\alpha}{\alpha+\beta}),\boldsymbol{M}}\left[\mathbb{E}_{\lambda\sim\mathrm{Beta}(\alpha+Q,\beta+1-Q)}\left[(1-Q)\ell(\tilde{\boldsymbol{Z}}_{v,v'},\boldsymbol{Y}_{v'})\right]\right]$$

(17)

$$= \frac{1}{m^2} \sum_{v,v'=1}^{m} \frac{\alpha}{\alpha + \beta} \mathbb{E}_{\lambda \sim \text{Beta}(\alpha+1,\beta),\boldsymbol{M}} \left[ \ell(\tilde{\boldsymbol{Z}}_{v,v'}, \boldsymbol{Y}_v) \right]$$

$$+ \frac{1}{m^2} \sum_{v,v'=1}^{m} \frac{\beta}{\alpha + \beta} \mathbb{E}_{\lambda \sim \text{Beta}(\alpha,\beta+1),\boldsymbol{M}} \left[ \ell(\tilde{\boldsymbol{Z}}_{v,v'}, \boldsymbol{Y}_{v'}) \right], \tag{18}$$

where Eq. (16) is due to Bernoulli distribution property and $\sum_{v,v'=1}^{m} \left[ \lambda \boldsymbol{Y}_v^\top (\tilde{\boldsymbol{Y}}_{v,v'} - \boldsymbol{Y}_v) + (1 - \lambda)\boldsymbol{Y}_{v'}^\top (\tilde{\boldsymbol{Y}}_{v,v'} - \boldsymbol{Y}_{v'}) \right] = \sum_{v,v'=1}^{m} \left[ \boldsymbol{Y}_v^\top (\tilde{\boldsymbol{Y}}_{v,v'} - \boldsymbol{Y}_v) \right] = (1 - \lambda) \sum_{v,v'=1}^{m} \left[ \boldsymbol{Y}_v^\top (\boldsymbol{Y}_{v'} - \boldsymbol{Y}_v) \right] = 0$, Eq. (17) holds due to conjugate property of beta and Bernoulli distribution, Eq. (18) is due to mean of $Q$ is $\alpha/(\alpha + \beta)$ when $Q = 1$, and mean of $1 - Q$ is $\frac{\beta}{\alpha+\beta}$ when $Q = 0$.

As $1 - \text{Beta}(\alpha, \beta + 1)$ and $\text{Beta}(\beta + 1, \alpha)$ are the same distribution and $\tilde{\boldsymbol{Z}}_{v,v'}(1 - \lambda) = \tilde{\boldsymbol{Z}}_{v',v}(\lambda)$, we have

$$\hat{\mathcal{L}}(f|\tilde{\boldsymbol{X}}, \tilde{\boldsymbol{Y}}, \tilde{\boldsymbol{A}}) = \frac{1}{m^2} \sum_{v,v'=1}^{m} \frac{\alpha}{\alpha + \beta} \mathbb{E}_{\lambda \sim \text{Beta}(\alpha+1,\beta),\boldsymbol{M}} \left[ \ell(\tilde{\boldsymbol{Z}}_{v,v'}, \boldsymbol{Y}_v) \right]$$

$$+ \frac{1}{m^2} \sum_{v,v'=1}^{m} \frac{\beta}{\alpha + \beta} \mathbb{E}_{\lambda \sim \text{Beta}(\beta+1,\alpha),\boldsymbol{M}} \left[ \ell(\tilde{\boldsymbol{Z}}_{v,v'}, \boldsymbol{Y}_v) \right]$$

$$= \frac{1}{m^2} \sum_{v,v'=1}^{m} \mathbb{E}_{\lambda' \sim \frac{\alpha}{\alpha+\beta}\text{Beta}(\alpha+1,\beta) + \frac{\beta}{\alpha+\beta}\text{Beta}(\beta+1,\alpha),\boldsymbol{M}} \left[ \ell(\tilde{\boldsymbol{Z}}_{v,v'}, \boldsymbol{Y}_v) \right]$$

$$= \frac{1}{m^2} \sum_{v,v'=1}^{m} \mathbb{E}_{\lambda' \sim \tilde{D}_\lambda,\boldsymbol{M}} \left[ \ell(\tilde{\boldsymbol{Z}}_{v,v'}, \boldsymbol{Y}_v) \right], \tag{19}$$

where Eq. (19) is due to $\tilde{D}_\lambda := \frac{\alpha}{\alpha+\beta}\text{Beta}(\alpha+1,\beta) + \frac{\beta}{\alpha+\beta}\text{Beta}(\beta+1,\alpha)$ and satisfying distribution property of $\tilde{D}_\lambda$. From the definition of $\boldsymbol{P}$ and $\check{\boldsymbol{Z}}_v$, we have

$$\hat{\mathcal{L}}(f|\tilde{\boldsymbol{X}}, \tilde{\boldsymbol{Y}}, \tilde{\boldsymbol{A}}) = \frac{1}{m} \sum_{v=1}^{m} \mathbb{E}_{\lambda' \sim \tilde{D}_\lambda,\boldsymbol{M}} \left[ \frac{1}{m} \sum_{v'=1}^{m} \ell(\tilde{\boldsymbol{Z}}_{v,v'}, \boldsymbol{Y}_v) \right]$$

$$= \frac{1}{m} \sum_{v=1}^{m} \mathbb{E}_{\lambda' \sim \tilde{D}_\lambda,\boldsymbol{M}} \left[ \mathbb{E}_{\boldsymbol{P}} \left[ \ell(\check{\boldsymbol{Z}}_v, \boldsymbol{Y}_v) \right] \right].$$

$\square$

**Lemma B.4.** *When $\tilde{D}_\lambda$ and $\boldsymbol{P}$ are the distribution of lambda and the permutation of mixed target samples defined in Lemma B.3. Then, we can approximate the empirical loss of GCN trained with iGraphMix by that of GCN trained without augmentation method as follows.*

$$\hat{\mathcal{L}}(f|\tilde{\boldsymbol{X}}, \tilde{\boldsymbol{Y}}, \tilde{\boldsymbol{A}}) \approx \hat{\mathcal{L}}(f|\boldsymbol{X}, \boldsymbol{Y}, \boldsymbol{A}) + R_1 + R_2,$$

*where* $R_1 := \frac{1}{m} \sum_{v=1}^{m} \mathbb{E}_{\lambda' \sim \tilde{D}_\lambda,\boldsymbol{M}} \left[ \mathbb{E}_{\boldsymbol{P}} \left[ \left( (\check{\boldsymbol{A}} + \boldsymbol{I})_v \check{\boldsymbol{H}}_1 - \boldsymbol{A}_v \boldsymbol{H}_1 \right) \boldsymbol{W}_2 (\boldsymbol{Z}_v - \boldsymbol{Y}_v) \right] \right]$

$R_2 := \frac{1}{m} \sum_{v=1}^{m} \mathbb{E}_{\lambda' \sim \tilde{D}_\lambda,\boldsymbol{M}} \left[ \mathbb{E}_{\boldsymbol{P}} \left[ \frac{1}{2} \boldsymbol{W}_2^\top \left( (\check{\boldsymbol{A}} + \boldsymbol{I})_v \check{\boldsymbol{H}}_1 - \boldsymbol{A}_v \boldsymbol{H}_1 \right)^\top \left( (\check{\boldsymbol{A}} + \boldsymbol{I})_v \check{\boldsymbol{H}}_1 - \boldsymbol{A}_v \boldsymbol{H}_1 \right) \boldsymbol{W}_2 \right] \right].$

*Proof.* Note that $\boldsymbol{Z}_v = \bar{\boldsymbol{A}}_v \boldsymbol{H}_1 \boldsymbol{W}_2$ and $\check{\boldsymbol{Z}}_v = (\check{\boldsymbol{A}} + \boldsymbol{I})_v \check{\boldsymbol{H}}_1 \boldsymbol{W}_2$ as shown in Lemma B.1. Then, we apply the second-order Taylor Theorem on $\ell\left( \check{\boldsymbol{Z}}_v, \boldsymbol{Y}_v \right)$ with respect to $\bar{\boldsymbol{A}}_v \boldsymbol{H}_1 \boldsymbol{W}_2$. We have

$$\ell(\check{\boldsymbol{Z}}_v, \boldsymbol{Y}_v) \approx \ell(\boldsymbol{Z}_v, \boldsymbol{Y}_v) + \left((\check{\boldsymbol{A}} + \boldsymbol{I})_v \check{\boldsymbol{H}}_1 \boldsymbol{W}_2 - \bar{\boldsymbol{A}}_v \boldsymbol{H}_1 \boldsymbol{W}_2\right) \left(\frac{\partial \ell(\boldsymbol{Z}_v, \boldsymbol{Y}_v)}{\partial \bar{\boldsymbol{A}}_v \boldsymbol{H}_1 \boldsymbol{W}_2}\right)$$
$$+ \frac{1}{2} \left((\check{\boldsymbol{A}} + \boldsymbol{I})_v \check{\boldsymbol{H}}_1 \boldsymbol{W}_2 - \bar{\boldsymbol{A}}_v \boldsymbol{H}_1 \boldsymbol{W}_2\right)^\top \frac{\partial^2 \ell(\boldsymbol{Z}_v, \boldsymbol{Y}_v)}{\partial (\bar{\boldsymbol{A}}_v \boldsymbol{H}_1 \boldsymbol{W}_2)^2} \left((\check{\boldsymbol{A}} + \boldsymbol{I})_v \check{\boldsymbol{H}}_1 \boldsymbol{W}_2 - \bar{\boldsymbol{A}}_v \boldsymbol{H}_1 \boldsymbol{W}_2\right).$$
$$(20)$$

Since $\ell(\boldsymbol{Z}_v, \boldsymbol{Y}_v) = \frac{1}{2} \|\boldsymbol{Z}_v - \boldsymbol{Y}_v\|^2$, we have the first-order and second-order term of $\ell$ as follows.

$$\begin{aligned} \frac{\partial \ell(\boldsymbol{Z}_v, \boldsymbol{Y}_v)}{\partial \bar{\boldsymbol{A}}_v \boldsymbol{H}_1 \boldsymbol{W}_2} &= \frac{\partial \boldsymbol{Z}_v}{\partial \bar{\boldsymbol{A}}_v \boldsymbol{H}_1 \boldsymbol{W}_2} \cdot \frac{\partial \ell(\boldsymbol{Z}_v, \boldsymbol{Y}_v)}{\partial \boldsymbol{Z}_v} \\ &= \boldsymbol{Z}_v - \boldsymbol{Y}_v, \end{aligned}$$
$$(21)$$

$$\begin{aligned} \frac{\partial^2 \ell(\boldsymbol{Z}_v, \boldsymbol{Y}_v)}{\partial (\bar{\boldsymbol{A}}_v \boldsymbol{H}_1 \boldsymbol{W}_2)^2} &= \frac{\partial}{\partial \bar{\boldsymbol{A}}_v \boldsymbol{H}_1} \left(\frac{\partial \ell(\boldsymbol{Z}_v, \boldsymbol{Y}_v)}{\partial \bar{\boldsymbol{A}}_v \boldsymbol{H}_1 \boldsymbol{W}_2}\right) \\ &= \frac{\partial}{\partial \bar{\boldsymbol{A}}_v \boldsymbol{H}_1 \boldsymbol{W}_2} \left(|\boldsymbol{Z}_v - \boldsymbol{Y}_v|\right) \\ &= \frac{\partial \boldsymbol{Z}_v}{\partial \bar{\boldsymbol{A}}_v \boldsymbol{H}_1 \boldsymbol{W}_2} \cdot \frac{\partial}{\partial \boldsymbol{Z}_v} \left(\boldsymbol{Z}_v - \boldsymbol{Y}_v\right) \\ &= \boldsymbol{I}. \end{aligned}$$
$$(22)$$

Substituting Eq. (21) and Eq. (22) to Eq. (20), we get

$$\ell(\check{\boldsymbol{Z}}_v, \boldsymbol{Y}_v) \approx \ell(\boldsymbol{Z}_v, \boldsymbol{Y}_v) + r_1 + r_2,$$

$$\begin{aligned} \text{s.t. } r_1 &= \left((\check{\boldsymbol{A}} + \boldsymbol{I})_v \check{\boldsymbol{H}}_1 - \bar{\boldsymbol{A}}_v \boldsymbol{H}_1\right) \boldsymbol{W}_2 (\boldsymbol{Z}_v - \boldsymbol{Y}_v), \\ r_2 &= \frac{1}{2} \left((\check{\boldsymbol{A}} + \boldsymbol{I})_v \check{\boldsymbol{H}}_1 \boldsymbol{W}_2 - \bar{\boldsymbol{A}}_v \boldsymbol{H}_1 \boldsymbol{W}_2\right)^\top \left((\check{\boldsymbol{A}} + \boldsymbol{I})_v \check{\boldsymbol{H}}_1 \boldsymbol{W}_2 - \bar{\boldsymbol{A}}_v \boldsymbol{H}_1 \boldsymbol{W}_2\right) \\ &= \frac{1}{2} \boldsymbol{W}_2^\top \left((\check{\boldsymbol{A}} + \boldsymbol{I})_v \check{\boldsymbol{H}}_1 - \bar{\boldsymbol{A}}_v \boldsymbol{H}_1\right)^\top \left((\check{\boldsymbol{A}} + \boldsymbol{I})_v \check{\boldsymbol{H}}_1 - \bar{\boldsymbol{A}}_v \boldsymbol{H}_1\right) \boldsymbol{W}_2. \end{aligned}$$

Thus, we obtain the result as follows.

$$\begin{aligned} \hat{\mathcal{L}}(f | \tilde{\boldsymbol{X}}, \tilde{\boldsymbol{Y}}, \tilde{\boldsymbol{A}}) &= \frac{1}{m} \sum_{v=1}^{m} \mathbb{E}_{\lambda' \sim \tilde{D}_\lambda, \boldsymbol{M}} \left[\mathbb{E}_{\boldsymbol{P}} \left[\ell(\check{\boldsymbol{Z}}_v, \boldsymbol{Y}_v)\right]\right] & (23) \\ &\approx \frac{1}{m} \sum_{v=1}^{m} \mathbb{E}_{\lambda' \sim \tilde{D}_\lambda, \boldsymbol{M}} \left[\mathbb{E}_{\boldsymbol{P}} \left[\ell(\boldsymbol{Z}_v, \boldsymbol{Y}_v) + r_1 + r_2\right]\right] \\ &= \frac{1}{m} \sum_{v=1}^{m} \mathbb{E}_{\lambda' \sim \tilde{D}_\lambda, \boldsymbol{M}} \left[\mathbb{E}_{\boldsymbol{P}} \left[\ell(\boldsymbol{Z}_v, \boldsymbol{Y}_v)\right]\right] \\ &\quad + \frac{1}{m} \sum_{v=1}^{m} \mathbb{E}_{\lambda' \sim \tilde{D}_\lambda, \boldsymbol{M}} \left[\mathbb{E}_{\boldsymbol{P}} \left[r_1\right]\right] + \frac{1}{m} \sum_{v=1}^{m} \mathbb{E}_{\lambda' \sim \tilde{D}_\lambda, \boldsymbol{M}} \left[\mathbb{E}_{\boldsymbol{P}} \left[r_2\right]\right] \\ &= \frac{1}{m} \sum_{v=1}^{m} \ell(\boldsymbol{Z}_v, \boldsymbol{Y}_v) + \frac{1}{m} \sum_{v=1}^{m} \mathbb{E}_{\lambda' \sim \tilde{D}_\lambda, \boldsymbol{M}} \left[\mathbb{E}_{\boldsymbol{P}} \left[r_1\right]\right] + \frac{1}{m} \sum_{v=1}^{m} \mathbb{E}_{\lambda' \sim \tilde{D}_\lambda, \boldsymbol{M}} \left[\mathbb{E}_{\boldsymbol{P}} \left[r_2\right]\right] \\ &= \mathcal{L}(f | \boldsymbol{X}, \boldsymbol{Y}, \boldsymbol{A}) + R_1 + R_2, \end{aligned}$$

where Eq. (23) is due to Lemma B.3. $\qquad \square$

Now, we are ready to prove Lemma 5.2. Remind Lemma B.4, we have

$$\hat{\mathcal{L}}(f | \tilde{\boldsymbol{X}}, \tilde{\boldsymbol{Y}}, \tilde{\boldsymbol{A}}) \approx \hat{\mathcal{L}}(f | \boldsymbol{X}, \boldsymbol{Y}, \boldsymbol{A}) + R_1 + R_2,$$

s.t. $R_1 = \sum_{v=1}^{m} \mathbb{E}_{\lambda', \boldsymbol{M}} \left[ \mathbb{E}_{\boldsymbol{P}} \left[ \left( (\boldsymbol{\check{A}} + \boldsymbol{I})_v \boldsymbol{\check{H}}_1 - \boldsymbol{\bar{A}}_v \boldsymbol{H}_1 \right)^\top \boldsymbol{W}_2^\top (\boldsymbol{Z}_v - \boldsymbol{Y}_v) \right] \right],$

$R_2 = \frac{1}{m} \sum_{v=1}^{m} \mathbb{E}_{\lambda', \boldsymbol{M}} \left[ \mathbb{E}_{\boldsymbol{P}} \left[ \frac{1}{2} \boldsymbol{W}_2^\top \left( (\boldsymbol{\check{A}} + \boldsymbol{I})_v \boldsymbol{\check{H}}_1 - \boldsymbol{\bar{A}}_v \boldsymbol{H}_1 \right)^\top \left( (\boldsymbol{\check{A}} + \boldsymbol{I})_v \boldsymbol{\check{H}}_1 - \boldsymbol{\bar{A}}_v \boldsymbol{H}_1 \right) \boldsymbol{W}_2 \right] \right],$

where $\lambda' \sim \frac{\alpha}{\alpha+\beta} \text{Beta}(\alpha+1, \beta) + \frac{\beta}{\alpha+\beta} \text{Beta}(\beta+1, \alpha)$.

Then, we should verify what $R_1$ and $R_2$ are. First, from Lemma B.2, we can get $R_1 = 0$.

For $R_2$, we have

$$R_2 = \frac{1}{m} \sum_{v=1}^{m} \mathbb{E}_{\lambda', \boldsymbol{M}} \left[ \mathbb{E}_{\boldsymbol{P}} \left[ \frac{1}{2} \boldsymbol{W}_2^\top \left( (\boldsymbol{\check{A}} + \boldsymbol{I})_v \boldsymbol{\check{H}}_1 - \boldsymbol{\bar{A}}_v \boldsymbol{H}_1 \right)^\top \left( (\boldsymbol{\check{A}} + \boldsymbol{I})_v \boldsymbol{\check{H}}_1 - \boldsymbol{\bar{A}}_v \boldsymbol{H}_1 \right) \boldsymbol{W}_2 \right] \right]$$

$$= \boldsymbol{W}_2^\top \frac{1}{2m} \sum_{v=1}^{m} \mathbb{E}_{\lambda', \boldsymbol{M}} \left[ \mathbb{E}_{\boldsymbol{P}} \left[ \left( (\boldsymbol{\check{A}} + \boldsymbol{I})_v \boldsymbol{\check{H}}_1 - \boldsymbol{\bar{A}}_v \boldsymbol{H}_1 \right)^\top \left( (\boldsymbol{\check{A}} + \boldsymbol{I})_v \boldsymbol{\check{H}}_1 - \boldsymbol{\bar{A}}_v \boldsymbol{H}_1 \right) \right] \right] \boldsymbol{W}_2$$

$$:= R(\boldsymbol{W}_2).$$

## B.2 PROOF OF THEOREM 5.3

Let $\boldsymbol{S} := 1/m \sum_{v=1}^{m} \mathbb{E}_{\lambda', \boldsymbol{M}} \left[ \mathbb{E}_{\boldsymbol{P}} \left[ \left( (\boldsymbol{\check{A}} + \boldsymbol{I})_v \boldsymbol{\check{H}}_1 - \boldsymbol{\bar{A}}_v \boldsymbol{H}_1 \right)^\top \left( (\boldsymbol{\check{A}} + \boldsymbol{I})_v \boldsymbol{\check{H}}_1 - \boldsymbol{\bar{A}}_v \boldsymbol{H}_1 \right) \right] \right] \in \mathbb{R}^{d_1 \times d_1}$. We reformulate $R(\boldsymbol{W}_2)$ for short-hand notation of the remaining proof.

$$R(\boldsymbol{W}_2) = \frac{1}{2} \boldsymbol{W}_2^\top \frac{1}{m} \sum_{v=1}^{m} \mathbb{E}_{\lambda', \boldsymbol{M}} \left[ \mathbb{E}_{\boldsymbol{P}} \left[ \left( (\boldsymbol{\check{A}} + \boldsymbol{I})_v \boldsymbol{\check{H}}_1 - \boldsymbol{\bar{A}}_v \boldsymbol{H}_1 \right)^\top \left( (\boldsymbol{\check{A}} + \boldsymbol{I})_v \boldsymbol{\check{H}}_1 - \boldsymbol{\bar{A}}_v \boldsymbol{H}_1 \right) \right] \right] \boldsymbol{W}_2$$

$$:= \frac{1}{2} \boldsymbol{W}_2^\top \boldsymbol{S} \boldsymbol{W}_2.$$

Next, we consider the constraint of $R(\boldsymbol{W}_2) \leq \omega$ where $\omega$ be the certain scalar value in Eq. (7). Then, the weight space $\boldsymbol{f}_{\text{iGraphMix}}$ is represented as follows.

$$\boldsymbol{f}_{\text{iGraphMix}} = \{ f \in \boldsymbol{f} | \|\boldsymbol{W}_1\|_\infty \leq \omega, \boldsymbol{W}_2^\top \boldsymbol{S} \boldsymbol{W}_2 \leq 2\omega \}. \tag{24}$$

Then, we introduce Transductive Rademacher Complexity (TRC) and the generalization error bound induced by TRC in Definition B.5 (El-Yaniv & Pechyony, 2009; Esser et al., 2021).

**Definition B.5** (Transductive Rademacher Complexity (El-Yaniv & Pechyony, 2009; Esser et al., 2021)). Suppose we have $m$ labeled samples over $n$ total samples, and $p \in [0, 0.5]$. Let $\mathcal{Z} \subset \mathbb{R}^n$ be the possible output logit space with $n$ samples and $\boldsymbol{\phi} = \{\phi_1, \cdots, \phi_n\}$ be the independent and identical distributed random vector where $\phi_i \pm 1$ with probability $2p$ and $\phi_i = 0$ with probability $1 - 2p$. Then, TRC is defined as follows.

$$\mathfrak{R}_{m,n}(\mathcal{Z}) := \left( \frac{1}{m} + \frac{1}{n-m} \right) \mathbb{E}_{\boldsymbol{\sigma}} \left[ \sup_{\boldsymbol{Z} \in \mathcal{Z}} \boldsymbol{\phi}^\top \boldsymbol{Z} \right]. \tag{25}$$

For any $f$ whose output logit $\boldsymbol{Z} \in \mathcal{Z}$ and the input graph ($\boldsymbol{X}$, $\boldsymbol{Y}$, and $\boldsymbol{A}$), the generalization error bound is bounded with TRC as follows.

$$\mathcal{L}(f|\boldsymbol{X}, \boldsymbol{Y}, \boldsymbol{A}) - \hat{\mathcal{L}}(f|\boldsymbol{X}, \boldsymbol{Y}, \boldsymbol{A}) \leq \mathfrak{R}_{m,n}(\mathcal{Z}) + O(n, m, \delta), \tag{26}$$

with probability $1 - \delta$ where $\mathcal{L}(f)$ and $\hat{\mathcal{L}}(f)$ be the generalization and empirical error of $f$.

Also, TRC has the scalar multiplication and convex hull property as in the below lemmas. Those lemmas are proved in Esser et al. (2021).

**Lemma B.6** (Linear combination). *Let $B \subset \mathbb{R}^n$, $C \subset \mathbb{R}^n$, and $t \in \mathbb{R}$. Then,*

*1) Scalar multiplication:* $\mathfrak{R}_{m,n}(\{t\boldsymbol{b} | \forall \boldsymbol{b} \in B\}) = \|t\|_1 \mathfrak{R}_{m,n}(B),$

*2) Addition:* $\mathfrak{R}_{m,n}(B + C) = \mathfrak{R}_{m,n}(B) + \mathfrak{R}_{m,n}(C).$

**Lemma B.7** (Convex hull). *Let $B \subset \mathbb{R}^n$, $B' = \{\sum_{j=1}^{N} \beta_j \boldsymbol{b}^{(j)} | N \in \mathbb{N}, \forall j, \boldsymbol{b}^{(j)} \in B, \beta_j \geq 0\}$, and $\boldsymbol{\beta} = [\beta_1, \ldots, \beta_N]$. Then,*

$$\mathfrak{R}_{m,n}(B) = \|\boldsymbol{\beta}\|_1 \mathfrak{R}_{m,n}(B').$$

Then, we need to find the TRC bound with output logit space induced by $\boldsymbol{f}_{\mathrm{iGraphMix}}$ to get the generalization error bound. Let $\boldsymbol{W}_2 := [[\boldsymbol{W}_2]_1, \cdots, [\boldsymbol{W}_2]_d]^\top \in \mathbb{R}^{d_1}$. Similar to the lemma given in Esser et al. (2021), we can get the second-layer TRC bound as follows.

**Lemma B.8** (Second-layer TRC bound). *Let $(\cdot)^{\dagger/2}$ be the squared inverse matrix given $(\cdot)$, and $\mathfrak{R}_{m,n}(\mathcal{H})$ be the TRC of the first-layer hidden representation space. Then, we get the second-layer output logit space TRC upper bound as follows.*

$$\mathfrak{R}_{m,n}(\mathcal{Z}) \leq 2\|\bar{\boldsymbol{A}}\|_\infty \sqrt{d_1}\omega \|\boldsymbol{S}^{\dagger/2}\|_\infty \mathfrak{R}_{m,n}(\mathcal{H}).$$

*Proof.* We first get the following inequality.

$$\begin{aligned}
\boldsymbol{Z}_v &= [\bar{\boldsymbol{A}}\boldsymbol{H}_1\boldsymbol{W}_2]_v \\
&= \left[ \sum_{l=1}^{d_1} [\boldsymbol{W}_2]_l [\bar{\boldsymbol{A}}\boldsymbol{H}_1]_{:,l} \right]_v \\
&= \left[ \sum_{l=1}^{d_1} [\boldsymbol{W}_2]_l \sum_{v'=1}^{n} \bar{\boldsymbol{A}}_{:,v'} [\boldsymbol{H}_1]_{:,l} \right]_v \\
&= \left[ \sum_{l=1}^{d_1} [\boldsymbol{W}_2]_l \left[ \sum_{v'=1}^{n} \bar{\boldsymbol{A}}_{:,v'} \right] [\boldsymbol{H}_1]_{:,l} \right]_v \\
&= \left[ \left[ \sum_{v'=1}^{n} \bar{\boldsymbol{A}}_{:,v'} \right] \sum_{l=1}^{d_1} [\boldsymbol{W}_2]_l [\boldsymbol{H}_1]_{:,l} \right]_v \\
&\leq \left[ \|\bar{\boldsymbol{A}}\|_\infty \sum_{l=1}^{d_1} \mathbf{1} \left[ [\boldsymbol{W}_2]_l [\boldsymbol{H}_1]_{:,l} \right] \right]_v ,
\end{aligned} \tag{27}$$

where Eq. (27) is due to the definition of $\|\cdot\|_\infty$.

From the above inequality, we define the new logit spaces $\mathcal{F}$ as follows.

$$\mathcal{F} = \left\{ \sum_{l=1}^{d_1} [\boldsymbol{W}_2]_l [\boldsymbol{H}_1]_{:,l} \,\middle|\, \boldsymbol{W}_2^\top \boldsymbol{S} \boldsymbol{W}_2 \leq 2\omega \right\}.$$

Since $\|\bar{\boldsymbol{A}}\|_\infty$ is a scalar, we apply Lemma B.6 to get the upper bound of $\mathfrak{R}_{m,n}(\mathcal{Z})$ with the above inequality as follows:

$$\mathfrak{R}_{m,n}(\mathcal{Z}) \leq \|\bar{\boldsymbol{A}}\|_\infty \mathfrak{R}_{m,n}(\mathcal{F}).$$

Let $\boldsymbol{V}_2 := \boldsymbol{S}^{1/2}\boldsymbol{W}_2 \in \mathbb{R}^{d_1}$ and $\dot{\boldsymbol{H}}_{:,l} := \boldsymbol{H}_{:,l}\boldsymbol{S}^{\dagger/2} \in \mathbb{R}^{n \times d_1}$. Then, we have the logit spaces $\dot{\mathcal{F}}, \dot{\mathcal{F}}_{\mathrm{inf}}$, and $\dot{\mathcal{F}}'_{\mathrm{inf}}$ as follows.

$$\dot{\mathcal{F}} = \left\{ \sum_{l=1}^{d_1} [\boldsymbol{V}_2]_l [\dot{\boldsymbol{H}}_1]_{:,l} \,\middle|\, \|\boldsymbol{V}_2\|_2 \leq 2\omega \right\},$$

$$\dot{\mathcal{F}}_{\mathrm{inf}} = \left\{ \sum_{l=1}^{d_1} [\boldsymbol{V}_2]_l [\dot{\boldsymbol{H}}_1]_{:,l} \,\middle|\, \|\boldsymbol{V}_2\|_1 \leq 2\sqrt{d_1}\omega \right\},$$

$$\dot{\mathcal{F}}'_{\mathrm{inf}} = \left\{ \sum_{l=1}^{d_1} [\boldsymbol{V}_2]_l [\dot{\boldsymbol{H}}_1]_{:,l} \,\middle|\, \|\boldsymbol{V}_2\|_1 = 2\sqrt{d_1}\omega \right\}.$$

Note that $\mathcal{F}$ and $\dot{\mathcal{F}}$ are equivalent. Since $\|\boldsymbol{V}_2\|_1 \leq \sqrt{d_1}\|\boldsymbol{V}_2\|_2$, we have $\dot{\mathcal{F}} \subset \dot{\mathcal{F}}_{\inf}$. Also, maximum value over $\boldsymbol{V}_2$ with inequality constraints is converged at the borderline, i.e. $\|\boldsymbol{V}_2\|_1 = 2\sqrt{d_1}\omega$. This induces $\mathfrak{R}_{m,n}(\dot{\mathcal{F}}_{\inf}) = \mathfrak{R}_{m,n}(\dot{\mathcal{F}}'_{\inf})$. Thus, we have

$$\mathfrak{R}_{m,n}(\mathcal{Z}) \leq \|\bar{\boldsymbol{A}}\|_\infty \mathfrak{R}_{m,n}(\mathcal{F})$$
$$= \|\bar{\boldsymbol{A}}\|_\infty \mathfrak{R}_{m,n}(\dot{\mathcal{F}})$$
$$\leq \|\bar{\boldsymbol{A}}\|_\infty \mathfrak{R}_{m,n}(\dot{\mathcal{F}}_{\inf})$$
$$= \|\bar{\boldsymbol{A}}\|_\infty \mathfrak{R}_{m,n}(\dot{\mathcal{F}}'_{\inf}).$$

Since we can think $V_2$ as the $\boldsymbol{\beta}$ in Lemma B.7,

$$\mathfrak{R}_{m,n}(\mathcal{Z}) \leq \|\bar{\boldsymbol{A}}\|_\infty \|\boldsymbol{V}_2\|_1 \mathfrak{R}_{m,n}(\dot{\mathcal{H}}'_{\inf})$$
$$\leq 2\|\bar{\boldsymbol{A}}\|_\infty \sqrt{d_1}\omega \mathfrak{R}_{m,n}(\dot{\mathcal{H}}'_{\inf}),$$

where $\dot{\mathcal{H}}'_{\inf}$ be the hidden logit space of $\dot{\boldsymbol{H}}_1$ as

$$\dot{\mathcal{H}}'_{\inf} = \left\{ \sum_{l=1}^{d_1} [\dot{\boldsymbol{H}}_1]_{:,l} \right\}.$$

Then, we get the following inequality.

$$\sum_{l=1}^{d_1} [\dot{\boldsymbol{H}}_1]_{:,l} = \sum_{l=1}^{d_1} \boldsymbol{H}_1 [\boldsymbol{S}^{\dagger/2}]_{:,l}$$
$$= \sum_{l'=1}^{d_1} \sum_{l=1}^{d_1} [\boldsymbol{H}_1]_{l'} [\boldsymbol{S}^{\dagger/2}]_{l',l}$$
$$= \sum_{l'=1}^{d_1} [\boldsymbol{H}_1]_{:,l} \sum_{l=1}^{d_1} [\boldsymbol{S}^{\dagger/2}]_{l',l}$$
$$\leq \sum_{l'=1}^{d_1} [\boldsymbol{H}_1]_{:,l'} \|[\boldsymbol{S}^{\dagger/2}]_{l'}\|_1$$
$$\leq \|\boldsymbol{S}^{\dagger/2}\|_\infty \sum_{l'=1}^{d_1} [\boldsymbol{H}_1]_{:,l'}.$$

Thus, from Lemma B.6, we have

$$\mathfrak{R}_{m,n}(\dot{\mathcal{H}}'_{\inf}) \leq \|\boldsymbol{S}^{\dagger/2}\|_\infty \mathfrak{R}_{m,n}(\mathcal{H}).$$

Therefore, we have the TRC upper bound as follows.

$$\mathfrak{R}_{m,n}(\mathcal{Z}) \leq 2\|\bar{\boldsymbol{A}}\|_\infty \sqrt{d_1}\omega \|\boldsymbol{S}^{\dagger/2}\|_\infty \mathfrak{R}_{m,n}(\mathcal{H}).$$

$\square$

Also, we get directly the first-layer TRC upper bound from Proposition 3 in Esser et al. (2021).

**Lemma B.9** (First-layer TRC bound (Esser et al., 2021)). *Let $\mathfrak{R}_{m,n}(\mathcal{H})$ be the TRC of the first-layer hidden representation space, and $L_\sigma$ be the Lipschitz constant. Then, we have the TRC upper bound of $\mathfrak{R}_{m,n}(\mathcal{H})$ as follows.*

$$\mathfrak{R}_{m,n}(\mathcal{H}) \leq L_\sigma \frac{n}{m(n-m)} \omega \|\bar{\boldsymbol{A}}\boldsymbol{X}\|_{2\to\infty} \sqrt{\frac{2\log n}{d_1}}.$$

Applying Lemma B.8 and Lemma B.9 to Eq. (26), we have

$$\mathcal{L}(f|\boldsymbol{X},\boldsymbol{Y},\boldsymbol{A}) - \hat{\mathcal{L}}(f|\boldsymbol{X},\boldsymbol{Y},\boldsymbol{A}) \leq \mathfrak{R}_{m,n}(\mathcal{Z}) + O(n,m,\delta)$$
$$\leq 2\|\bar{\boldsymbol{A}}\|_\infty \sqrt{d_1}\omega \|\boldsymbol{S}^{\dagger/2}\|_\infty \mathfrak{R}_{m,n}(\mathcal{H}) + O(n,m,\delta)$$
$$\leq 2L_\sigma \frac{n}{m(n-m)} \|\bar{\boldsymbol{A}}\|_\infty \sqrt{d_1}\omega \|\boldsymbol{S}^{\dagger/2}\|_\infty \omega \|\bar{\boldsymbol{A}}\boldsymbol{X}\|_{2\to\infty} \sqrt{\frac{2\log n}{d_1}} + O(n,m,\delta)$$
$$= \frac{n}{m(n-m)} abc \|\bar{\boldsymbol{A}}\|_\infty \|\bar{\boldsymbol{A}}\boldsymbol{X}\|_{2\to\infty} \sqrt{\log n} + O(n,m,\delta),$$

where $a = 2L_\sigma\omega$, $b = \omega\sqrt{2/d_1}$, and $c = \sqrt{d_1}\|S^{\dagger/2}\|_\infty$. Note that, in the definition of $S$, $S$ is a function of $\lambda'$, $\check{A}$, and $\check{H}_1$. This means that $c$ is the function of $\alpha$, $A$, $X$. Therefore, $c = Q(\alpha, A, X)$ where $Q$ is the certain function, then Theorem 5.3 is satisfied.

## C EXPERIMENTAL SETTINGS

### C.1 DATASETS

We used CiteSeer, CORA, PubMed, ogbn-arxiv, and Flickr for the transductive setting. The detailed statistics for each dataset are summarized in Table 3.

Table 3: Datasets statistics for the transductive setting.

| Dataset | CiteSeer | CORA | PubMed | ogbn-arxiv | Flickr |
|---|---|---|---|---|---|
| # Nodes | 3,327 | 2,708 | 19,717 | 169,343 | 89,250 |
| # Edges | 4,552 | 5,278 | 88,648 | 1,166,243 | 899,756 |
| # Features | 3,703 | 1,433 | 500 | 128 | 500 |
| # Class | 6 | 7 | 3 | 40 | 7 |
| # Valid Nodes | 500 | 500 | 500 | 29,799 | 22,313 |
| # Test Nodes | 1,000 | 1,000 | 1,000 | 48,603 | 22,313 |
| Sparsity | 0.082 | 0.144 | 0.023 | 0.008 | 0.011 |
| Average Degree | 2.74 | 3.90 | 4.50 | 2.74 | 10.08 |
| Edges Between Labeled Nodes Ratio (%) | 1.78 | 0.40 | 0.00 | 0.32 | 0.21 |

**Planetoid Benchmark** We considered the three benchmark citation network datasets for the transductive node classification task: CiteSeer, CORA, and PubMed (Sen et al., 2008). The goal of the node classification of these datasets is to predict the appropriate subject class for each paper. CiteSeer, CORA, and PubMed contain scientific publications, machine learning publications, and biomedical publications respectively. Thus, they consist of the papers as the nodes, citation links between papers as the edges, and the subject of the papers as the labels. Node features are constructed via the bag-of-words feature representation of the paper document. We followed the labeled node per class and the train/test dataset split settings for Table 1 used in Yang et al. (Yang et al., 2016).

**Open Graph Benchmark (OGB)** We also considered the one citation network dataset for the transductive node classification task: ogbn-arxiv (Hu et al., 2020). The goal of the node classification of those datasets is to predict the appropriate subject class for each paper. ogbn-arxiv contains all computer science (CS) arxiv papers provided by Hu et al. (Hu et al., 2020). Thus, it consists of the papers as the nodes, citation links between papers as the edges, and the subject of the papers as the labels. Node features are constructed via the skip-gram representation of papers' titles and abstracts. To make ogbn-arxiv semi-supervised, we sampled 10% nodes from the original training node set for Table 1 (Fang et al., 2023).

**Flickr** We considered one social network dataset for the transductive node classification task: Flickr (McAuley & Leskovec, 2012). The goal of the dataset is to predict the category and the community of each post. The nodes for Flickr is the image posts. The edges between nodes exist if the images share common properties, e.g., geometric location, gallery, etc. Node features are the image pixels. The training and evaluation nodes are split by creating dates of post (Hamilton et al., 2017). For Flickr, we used the original training node sets, similar to Fang et al. (2023) for Table 1.

### C.2 BACKBONE ARCHITECTURE

To evaluate the effect of iGraphMix in various GNNs, we utilized three well-known backbone GNNs. First, we used GCN (Kipf & Welling, 2017), which is the simple version of the diffusion operator in GNNs. Second, we utilized GAT (Veličković et al., 2018) and GATv2 (Brody et al., 2022), which use attention mechanisms in the diffusion operator in GNNs.

### C.3 HYPERPARAMETERS

Hyperparameter settings of all datasets and models are as follows. For CiteSeer, CORA, and Pubmed, we used Adam Optimizer (Kingma & Ba, 2015) with 0.01 learning rate and 5e-4 weight decaying,

dropout with 0.5 probability, and 16 hidden units for GCN. Also, we used Adam Optimizer with a learning rate of 0.005 and weight decaying of 5e-4, dropout of 0.5 probability, 16 hidden units, and 1 head for GAT and GATv2 (Zhao et al., 2021; Verma et al., 2021). We trained the above models by 2000 epochs and reported the test scores when the validation scores were the maximum. For ogbn-arxiv, we used Adam Optimizer (Kingma & Ba, 2015) with 0.005 learning rate and 5e-4 weight decaying, dropout with 0.0 probability, 256 hidden units, and 3 number of layers for GCN, GAT, and GATv2 (Fang et al., 2023). Also, we used 1 head for GAT and GATv2. We added batch normalization layers between GNN layers. We trained the above models by 200 epochs and reported the test scores when the validation scores are the maximum. For Flickr, we used Adam Optimizer (Kingma & Ba, 2015) with 0.001 learning rate and 5e-5 weight decaying, dropout with 0.0 probability, 256 hidden units, and 2 number of layers for GCN, GAT, and GATv2 (Fang et al., 2023). Also, we used 1 head for GAT and GATv2. We trained the above models by 2000 epochs and reported the test scores when the validation scores were the maximum.

### C.4 GRAPH DATA AUGMENTATION METHODS

For the baseline augmentation methods, we found the best drop probability parameters from $[0.1, 0.9]$ for DropEdge (Rong et al., 2020), DropNode (Feng et al., 2020), and DropMessage (Fang et al., 2023), and beta distribution probability from $\{0.001, 0.005, 0.01, 0.05, 0.1, 0.5, 1, 5, 10, 50, 100\}$ for M-Mixup (Wang et al., 2021), and iGraphMix.

### C.5 EXPERIMENTAL DETAILS

We conducted our experiments on the V-100 with CUDA version 11.3. GNNs for our experiments are built on Pytorch [1] Paszke et al. (2019) with version 1.12.1. and Pytorch Geometric [2] (Fey & Lenssen, 2019) with version 2.1.0. For the baseline augmentation methods, we used the implementation of DropEdge (Rong et al., 2020) and DropNode (Feng et al., 2020) in Pytorch Geometric and employed the original paper implementation for DropMessage (Fang et al., 2023) [3]. For M-Mixup (Wang et al., 2021), we utilized the original paper implementation [4].

## D ADDITIONAL RESULTS

We reported the additional results that were not handled in the main context due to the space limitation.

### D.1 GENERALIZATION GAP

**Compared with Baselines**  We reported the generalization gap analysis for GCN on the other transductive datasets: CORA, PubMed, Flickr, and ogbn-arxiv. Figure 5 showed the generalization gap results on CORA, PubMed, Flickr, and ogbn-arxiv, similar to Figure 2. We could confirm a consistently smaller generalization gap in iGraphMix compared to no augmented training. This result may support our theoretical claim in Section 5 that iGraphMix showed a smaller generalization gap compared to standard training with high probability. Furthermore, we found that the final generalization gap of iGraphMix tends to be consistently smaller than other augmentation baseline methods in these datasets. This can empirically support the claim in Section 6.2 that iGraphMix does make a smaller generalization gap not only in CiteSeer but also in other datasets, such as CORA, PubMed, Flickr, and ogbn-arxiv.

**Beta distribution parameters** $\alpha$  We observed the generalization gap with respect to $\alpha$ on CORA, and Pubmed for 2-layer GCN in Figure 6. Similar to the results in Section 6.2, the generalization gap of iGraphMix in CORA and PubMed becomes smaller when $\alpha$ is large. Notably, we verified that the optimal $\alpha$ of these datasets is 50. On the other hand, as shown in Figure 3, the optimal $\alpha$ of CiteSeer, ogbn-arxiv, and Flickr are 100, 10 and 50, respectively. Therefore, these results may strengthen the claim in Section 6.2 that the different datasets require different optimal $\alpha$ to achieve the small generalization gap.

---

[1]`https://github.com/pytorch/pytorch`
[2]`https://github.com/pyg-team/pytorch_geometric`
[3]`https://github.com/zjunet/DropMessage`
[4]`https://github.com/vanoracai/MixupForGraph`

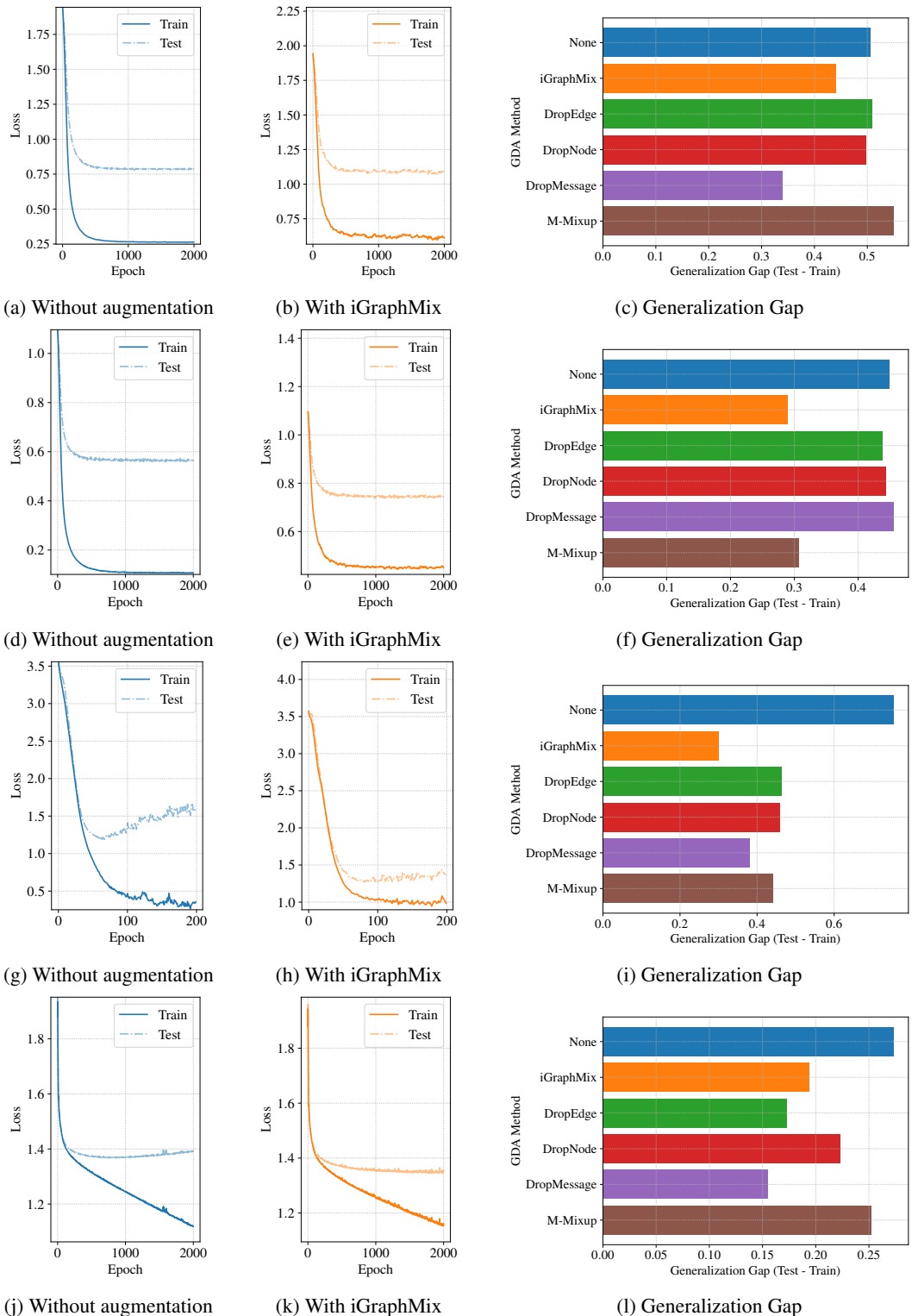

Figure 5: Generalization Results of GCN with CORA ((a)-(c)), PubMed ((d)-(e)), ogbn-arxiv ((g)-(h)), and Flickr ((j)-(l)).

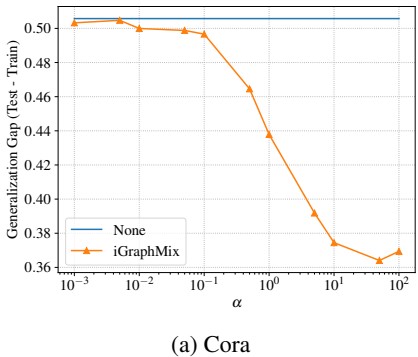

(a) Cora

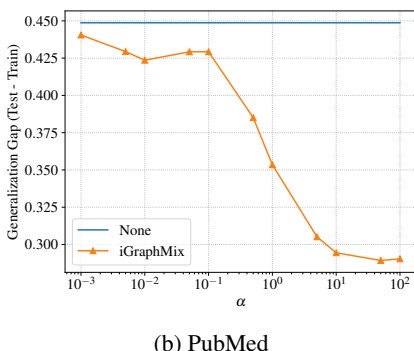

(b) PubMed

Figure 6: Generalization gap results on other datasets with respect to Beta distribution's parameter $\alpha$ for 2-layer GCN. A small generalization gap reveals a better result.

Table 4: Ablation results on three transductive datasets.

| Augmentation Methods | CiteSeer | Cora | PubMed |
|---|---|---|---|
| None | 72.05 (0.56) | 82.65 (0.55) | 79.32 (0.15) |
| iGraphMix (input+label) | 71.50 (0.47) | 81.26 (0.27) | 79.18 (0.53) |
| iGraphMix (edge+label) | 72.44 (0.81) | 82.72 (0.21) | 79.18 (0.33) |
| iGraphMix (input+edge+label) (*proposed*) | **73.67 (0.61)** | **83.78 (0.42)** | **79.93 (0.60)** |

## D.2 ANALYSIS ON VARIOUS SETTINGS

We assessed the performance of GCN with various settings, such as the various number of layers and labeled nodes on CORA, PubMed, ogbn-arxiv, and Flickr, which we did not describe in the main context due to paper limitations. Figure 7 showed the micro-F1 score of the various number of layers and the number of labeled nodes on CORA, PubMed, ogbn-arxiv, and Flickr with the 2-layer GCN model. Note that we considered the ratio of labeled samples rather than the number of labeled nodes per class on ogbn-arxiv and Flickr. This is because these datasets originally provided labeled nodes based on the overall labeled ratio, not the per-class ratio.

From Figures 7a, 7c, 7e, and 7g, we observed a consistent improvement of the micro-F1 score in iGraphMix with varying numbers of layers on these datasets, similar to the result on CiteSeer shown in Figure 4a. These results may support our claim that iGraphMix improves the micro-F1 score in the various number of layers. Additionally, these results may imply that the proposed method mitigates the over-smoothing problem.

Moreover, Figures 7b, 7d, 7f, and 7h showed that the proposed method improves the micro-F1 score regardless of the number of nodes. Importantly, we found that our method can be more effective in improving the micro-F1 score when training with fewer $L$ than fully labeled nodes on these datasets, like the result on CiteSeer shown in Figure 4b. This result may also imply that our method addresses the over-fitting problem, especially with fewer labeled nodes.

## D.3 ABLATION STUDY

We conducted the ablation study on iGraphMix and the result is summarized in Table 4. We compared our method with 1) applying mixup only to input and label, iGraphMix (input+label), and 2) applying mixup only to edge and label, iGraphMix (edge+label), on three transductive datasets, CiteSeer, Cora, and PubMed, for node classification. We observed that iGraphMix (input + edge+ label) consistently outperforms not only the standard training but also iGraphMix (input+label), and iGraphMix (edge+label). This result may imply that interpolating all components of the graph is crucial, showing our method's effectiveness. Furthermore, iGraphMix (input+label) consistently underperforms the standard training, and iGraphMix (edge+label) is better or comparable to the standard training. It may suggest the importance of applying mixup on the edge for graph datasets.

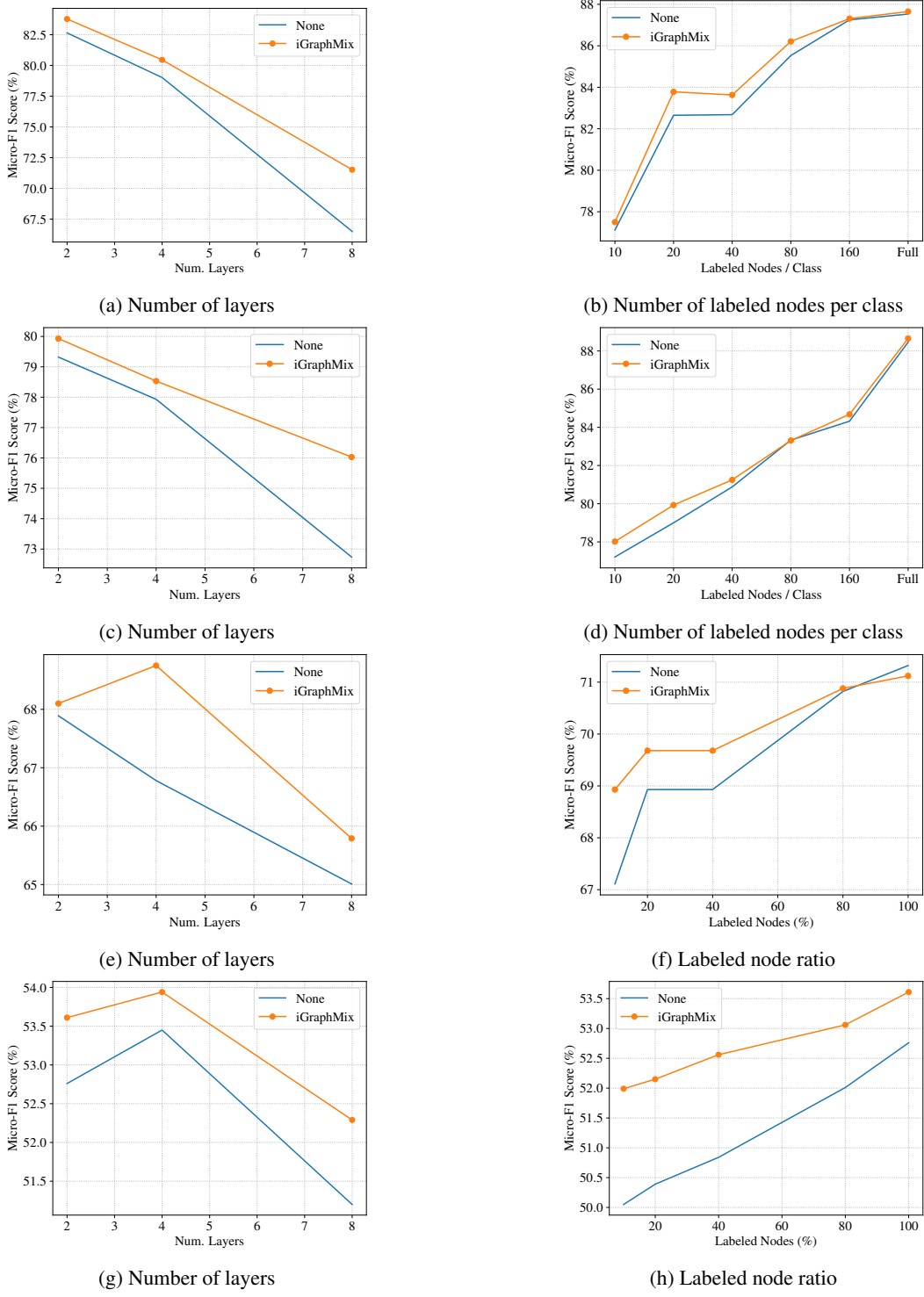

Figure 7: Micro-F1 score in various different number of layers and labeled node ratio on Cora((a)-(b)), PubMed((c)-(d)), ogbn-arxiv((e)-(f)), and Flickr((g)-(h)) datasets. The results are the average scores of 10 trials.

Table 5: Robustness results with feature noise on three transductive datasets.

| Dataset | CiteSeer | | | Cora | | | PubMed | | |
|---|---|---|---|---|---|---|---|---|---|
| Noise level (%) | 10% | 20% | 40% | 10% | 20% | 40% | 10% | 20% | 40% |
| None | 66.90 | 66.89 | 66.12 | 78.10 | 77.76 | 77.92 | 75.46 | 74.62 | 73.36 |
| iGraphMix | 68.48 | 68.54 | 68.30 | 79.00 | 78.64 | 78.76 | 75.78 | 75.80 | 74.88 |

Table 6: Robustness results with structural noise on three transductive datasets.

| Dataset | CiteSeer | | | Cora | | | PubMed | | |
|---|---|---|---|---|---|---|---|---|---|
| Noise level (%) | 10% | 20% | 40% | 10% | 20% | 40% | 10% | 20% | 40% |
| None | 67.62 | 61.88 | 54.90 | 78.62 | 73.14 | 60.96 | 73.92 | 71.88 | 60.20 |
| iGraphMix | 68.94 | 63.22 | 55.82 | 79.14 | 74.06 | 61.78 | 74.54 | 72.56 | 61.22 |

Table 7: Robustness results with label noise on three transductive datasets.

| Dataset | CiteSeer | | | Cora | | | PubMed | | |
|---|---|---|---|---|---|---|---|---|---|
| Noise level (%) | 10% | 20% | 40% | 10% | 20% | 40% | 10% | 20% | 40% |
| None | 68.24 | 64.14 | 57.72 | 79.10 | 76.00 | 67.60 | 77.32 | 75.76 | 68.64 |
| iGraphMix | 69.08 | 65.92 | 58.62 | 80.02 | 76.42 | 69.08 | 77.66 | 76.00 | 69.04 |

## D.4 ROBUSTNESS TEST

We evaluated the robustness of our method on three noise settings: 1) feature noise, 2) structural noise, and 3) label noise. We followed the noise setting used in Fang et al. (2023); the feature noise is provided by adding standard Gaussian noise to input features with a probability of noise level; the structural noise is given by randomly adding and removing edges with a probability of noise level; for label noise, we randomly permuted training labels with a probability of noise level for this experiment. We used GCN as the backbone model for all settings and assessed the robustness of Cora, CiteSeer, and PubMed. The results are summarized below.

**Feature Noise**   We evaluated the robustness when the feature noise is introduced in the graph. The feature noise is provided by adding standard Gaussian noise to input features with a probability of noise level. Table 5 shows how robust our method is for the feature noise. We indicated that our method outperforms the standard training in all noise levels.

**Structural Noise**   We assessed the robustness when the graph contains structural noise. The structural noise is given by randomly adding and removing edges with a probability of noise level. Table 6 shows the results of how the proposed method is robust to the structural noise. We found that when structural noise is introduced, GCN trained with iGraphMix outperforms GCN trained without iGraphMix (None). In addition, as the noise level increases from 10% to 40%, our method achieves more performance improvement compared to the standard training (None).

**Label Noise**   We assessed the robustness when the graph contains label noise. We randomly permuted training labels with a probability of noise level for this experiment. We trained GCN with the noisy training labels and evaluated GCN's performance. Table 7 shows how robust our method for noise labels is. We indicated that our method outperforms the standard training in all noise levels.

## D.5 COMPUTATIONAL COMPLEXITY

We assessed the computational complexity of our method by Big O complexity analysis. Then, we verified the computational complexity from the experiments on Cora and ogbn-arxiv.

Table 8 shows the time complexity with the Big O notation of all methods including five baselines and the proposed method. Specifically, let $\mathcal{V}$ be the number of nodes, $\mathcal{E}$ be the number of total edges, $\mathcal{E}_{\text{tr}}$ be the number of edges connected to training nodes, $d$ be the hidden dimension of the GNN layer, and $K$ be the number of layers. Then, we split the complexity into two parts, i.e., augmented data preparation and graph convolution, and get the Big O complexity of the parts as follows. The time

Table 8: Big O complexity for all data augmentation methods in GCN. We split the data augmentation process into 1) augmented data preparation and 2) graph convolution.

| Operation | None | DropEdge | DropNode | DropMessage | M-Mixup | iGraphMix |
|---|---|---|---|---|---|---|
| 1) Data Preparation | $\mathcal{O}(1)$ | $\mathcal{O}(|\mathcal{E}|)$ | $\mathcal{O}(|\mathcal{E}|+|\mathcal{V}|)$ | $\mathcal{O}(1)$ | $\mathcal{O}(|\mathcal{E}|+2|\mathcal{V}|)$ | $\mathcal{O}(|\mathcal{E}|+|\mathcal{V}|)$ |
| 2) Graph Convolution | $\mathcal{O}\left(|\mathcal{E}|d^K\right)$ | $\mathcal{O}\left(p|\mathcal{E}|d^K\right)$ | $\mathcal{O}\left(p|\mathcal{E}|d^K\right)$ | $\mathcal{O}\left(|\mathcal{E}|d^K\right)$ | $\mathcal{O}\left(2|\mathcal{E}|d^K\right)$ | $\mathcal{O}\left(|\mathcal{E}|d^K\right)$ |

Table 9: Time complexity of all data augmentation methods in GCN on benchmark datasets.

| Dataset | Operation | None | DropEdge | DropNode | DropMessage | M-Mixup | iGraphMix |
|---|---|---|---|---|---|---|---|
| Cora | 1) Data Preparation | 0.83e-3 | 1.13e-3 | 1.33e-3 | **0.83e-3** | 1.96e-3 | 3.72e-3 |
| | 2) Graph Convolution | 1.12e-2 | **1.10e-2** | 1.10e-2 | 1.11e-2 | 1.73e-2 | 1.57e-2 |
| ogbn-arxiv | 1) Data Preparation | **1.25e-3** | 1.34e-3 | 1.39e-3 | **1.25e-3** | 2.61e-2 | 2.56e-2 |
| | 2) Graph Convolution | 3.32e-2 | **1.76e-2** | 1.81e-2 | 3.30e-2 | 8.33e-2 | 3.37e-2 |

complexity of GCN (None) was represented in Veličković et al. (2018). Since there are dropping edges or nodes in DropEdge and DropNode, the complexity for data preparation of DropEdge and DropNode is $\mathcal{O}(|\mathcal{E}|)$ and $\mathcal{O}(|\mathcal{E}|+|\mathcal{V}|)$, respectively. Also, the complexity of graph convolution is $\mathcal{O}\left(p|\mathcal{E}|d^K\right)$, where $p \in [0,1]$ is a dropping probability. DropMessage considers all edges and nodes in all layers, their complexity is the same as the standard GCN. For M-Mixup, the complexities for data preparation and graph convolution are $\mathcal{O}(|\mathcal{E}|+2|\mathcal{V}|)$ and $\mathcal{O}\left(2|\mathcal{E}|d^K\right)$, respectively. Lastly, in the case of iGraphMix, the complexities for data preparation is $\mathcal{O}(|\mathcal{E}|+|\mathcal{V}|)$ and that for graph convolution is $\mathcal{O}\left(|\mathcal{E}|d^K\right)$.

Table 9 shows the computational time of data preparation and graph convolution on Cora and ogbn-arxiv datasets. The below table shows the mean computational time (second) per epoch. We could find that the real computational time is consistent with our Big O complexity calculation.

From the analysis, we found that our method would not need lots of computational complexity, especially compared to M-Mixup. Although some methods require less computational time, our method leads to better performance when training GNNs, highlighting the necessity of the proposed method.

# E    EXPERIMENTS ON OTHER TASKS

In this section, we provided the details and results of other graph tasks, such as inductive node classification, and link prediction.

## E.1    INDUCTIVE NODE CLASSIFICATION

### PROBLEM DEFINITION

The inductive setting states that we train the model only using the sub-graph with training nodes and edges between them and evaluate the model using the sub-graph with evaluation nodes and edges between them. Since the inductive setting is considered to handle large-scale graphs in general, methods for this setting are mainly focused on sub-graph sampling. For instance, GraphSAGE (Hamilton et al., 2017) randomly samples neighboring nodes for each node. GraphSAINT (Zeng et al., 2020) splits the original graphs into several sub-graphs, and randomly samples sub-graphs from the original graphs. Other works proposed various sub-graph sampling methods (Chen et al., 2018b;a; Chiang et al., 2019).

### DATASETS

We used Reddit and ogbn-products for the inductive node classification. The detailed statistics for each dataset are summarized in Table 10.

**Reddit**    We considered one social network dataset: Reddit (Grover & Leskovec, 2016). The goal of this dataset is to predict the category and the community of each post. The nodes for Reddit are the text posts. The edges are connected if the same users interact with both posts. Node features are GloVE CommonCrawl word vectors (Pennington et al., 2014) of the post for Reddit. The training and evaluation nodes are split by creating dates of post (Hamilton et al., 2017). To make the inductive

setting semi-supervised, we sampled 10% nodes from the original training node set (Hamilton et al., 2017; Zeng et al., 2020).

**Open Graph Benchmarks (OGB)**     We also considered one e-commerce network dataset: ogbn-products (Hu et al., 2020). The goal of this dataset is to predict the category of the products. The nodes for ogbn-products are the products' descriptions. The edges are connected if two products are purchased at the same time. Node features are constructed via bag-of-words and applied Principal Component Analysis to reduce the dimensions of node features. The training and evaluation nodes are split by sales ranking. To make ogbn-products semi-supervised, we sampled 10% nodes from the original training node-set.

Table 10: Datasets statistics for the inductive setting

| Dataset | Reddit | ogbn-products |
|---|---|---|
| # Nodes | 232,965 | 2,449,029 |
| # Edges | 114,615,892 | 61,859,140 |
| # Features | 602 | 100 |
| # Class | 41 | 47 |
| # Valid Nodes | 23,699 | 244,903 |
| # Test Nodes | 55,334 | 244,903 |

BACKBONE MODELS

Since one of our goals is to show the effect of iGraphMix in the various sampling methods in the inductive setting, we considered two well-known subgraph sampling methods. We utilized Random Walk sampling used in GraphSAGE (Hamilton et al., 2017) and Sub-graph sampling used in GraphSAINT (Zeng et al., 2020). For the base model architecture, we used GraphSAGE (Hamilton et al., 2017) layers for GraphSAGE and GCN with JK-Nets (Xu et al., 2018) for GraphSAINT. For both models, we used Adam Optimizer with a 0.001 learning rate and 2e-4 weight decaying for Reddit. We used dropout with 0.2 probability. We also utilized 128 hidden units (Zeng et al., 2020). The maximum epochs are 200 for GraphSAGE and 50 for GraphSAINT. Similar to the transductive setting, we reported the test score when the validation score is the maximum. The other detailed parameters for sampling are followed by GraphSAGE (Hamilton et al., 2017) and GraphSAINT (Zeng et al., 2020).

RESULTS

Although we do not provide the theoretical results on the inductive setting, we experimentally showed that iGraphMix helps model training even in this setting. The results for the inductive setting were represented in Table 11. We could find that training GNNs with iGraphMix tends to outperform that without augmentation methods on various datasets and backbone architectures. Furthermore, we could observe that iGraphMix shows a performance improvement comparable to other augmentation methods. These results verified that the proposed method could improve the performance not only in the transductive small graphs but also in the large inductive graphs.

E.2   LINK PREDICTION

PROBLEM DEFINITION

Link prediction is a task in graph analysis that predicts whether two nodes are connected or not. For mathematical notation, we denote the output of GNNs $\boldsymbol{Z} \in \mathbb{R}^{n \times n}$ as follows. $\boldsymbol{Z} = \text{sigmoid}\left(\boldsymbol{H}^K \cdot \boldsymbol{H}^{K^\top}\right)$, where $\boldsymbol{H}^K \in \mathbb{R}^{n \times d_K}$ is the hidden representation matrix of GNNs with feature dimension $d_K$.

As represented in Verma et al. (2021), link prediction can be represented as a form of two-class node classification. Specifically, when a new graph $\mathcal{G}'$ is generated by combining the input features and the edges of two pairs of nodes from the original graph $\mathcal{G}$, the goal of link prediction becomes predicting the class of nodes in $\mathcal{G}'$. These classes indicate whether the two pairs of nodes are connected or not in $\mathcal{G}$. This process treats the link prediction in $\mathcal{G}$ as the node classification in $\mathcal{G}'$.

Table 11: Overall Micro-F1 score (%) on inductive datasets when the number of layers is two. The results are the average scores and standard deviations with different random seeds. Since M-Mixup showed out-of-memory for GraphSAGE and ogbn-products, we represent the result as OOM.

| Backbone | Data Augmentation | Datasets | |
|---|---|---|---|
| | | Reddit | ogbn-products |
| GraphSAGE | None | 94.87 (0.03) | 74.41 (0.61) |
| | DropEdge | 94.91 (0.05) | **75.21 (0.17)** |
| | DropNode | 95.31 (0.03) | 74.98 (0.19) |
| | DropMessage | 94.86 (0.04) | 74.70 (0.26) |
| | M-Mixup | OOM | OOM |
| | iGraphMix (*ours*) | **95.33 (0.04)** | 75.04 (0.14) |
| GraphSAINT | None | 89.13 (0.25) | 50.41 (0.35) |
| | DropEdge | 89.30 (0.29) | **56.02 (0.09)** |
| | DropNode | 89.26 (0.24) | 52.53 (0.18) |
| | DropMessage | 89.18 (0.27) | 50.22 (0.48) |
| | M-Mixup | **89.91 (0.32)** | OOM |
| | iGraphMix (*ours*) | 89.84 (0.33) | 53.49 (0.23) |

DATASETS

We used CiteSeer, CORA, and PubMed for the link prediction task.

**Planetoid Benchmark**    We considered the three benchmark citation network datasets for the link prediction task: CiteSeer, CORA, and PubMed. The goal of the link prediction of these datasets is to predict the existence of citation links between every two papers. The detailed statistics of the number of nodes and edges of these datasets were summarized in Table 3. Followed by Fey & Lenssen (2019), we sampled 5% and 10% edges from the original edges for valid and test edges respectively, and the remaining edges were used for training.

IGRAPHMIX FOR LINK PREDICTION

For link prediction, iGraphMix generates virtual graphs and labels in a batch-wise manner as in Definition E.1. The difference between iGraphMix on node classification and link prediction is the second equation of Eq. (28). While iGraphMix linearly interpolates two one-hot label matrices on node classification, the proposed method on link prediction generates a virtual label matrix for edges by masking the label matrices and aggregating them. Specifically, our method first masks the label matrices for edges using the masking matrices that are the same in the last equation of Eq. (28). Then, the proposed method adds the two masked label matrices to generate the virtual label matrix used for training.

**Definition E.1** (iGraphMix for link prediction). Let $\boldsymbol{M}_\lambda$ be the masking matrix with $\lambda$ dropping probability. iGraphMix mixes feature matrix, label matrix, and adjacency matrix as follows:

$$
\begin{aligned}
\tilde{\boldsymbol{X}} &:= \lambda \boldsymbol{X} + (1 - \lambda)\, \boldsymbol{X}', \\
\boldsymbol{A}^{\tilde{Y}} &:= \boldsymbol{M}_{1-\lambda} \circ \boldsymbol{A}^Y + \boldsymbol{M}_\lambda \circ \boldsymbol{A}^{Y'}, \\
\tilde{\boldsymbol{A}} &:= \boldsymbol{M}_{1-\lambda} \circ \boldsymbol{A} + \boldsymbol{M}_\lambda \circ \boldsymbol{A}',
\end{aligned}
\tag{28}
$$

where $(\boldsymbol{X}', \boldsymbol{A}^{Y'}, \boldsymbol{A}')$ is the permuted batch within labeled nodes of $(\boldsymbol{X}, \boldsymbol{A}^Y, \boldsymbol{A})$ .

RESULTS

We experimentally assessed that iGraphMix helps GNNs improve performance on link prediction. We conducted the link prediction task on CiteSeer, Cora, and PubMed datasets and evaluated the performance of iGraphMix with other augmentation methods in terms of ROC-AUC. For the base model architecture, we used a two-layer GCN with 128 hidden units. Then, we used Adam Optimizer with a 0.01 learning rate for all datasets, and the maximum epochs were 100. The results of GCN on link prediction were presented in Table 12. Similar to the other tasks, we reported the test ROC-AUC when the ROC-AUC of the validation sets is the maximum. We found that the proposed method significantly outperforms baseline methods on two datasets out of three. Although our method showed

Table 12: Overall results on link prediction task.

| Backbone | Data Augmentation | Link Prediction | | |
|---|---|---|---|---|
| | | CiteSeer (ROC-AUC) | Cora (ROC-AUC) | PubMed (ROC-AUC) |
| GCN | None | 93.22 (0.33) | 90.08 (0.54) | 97.36 (0.06) |
| | DropEdge | 94.49 (0.31) | 91.94 (0.84) | 97.55 (0.09) |
| | DropNode | 93.30 (0.40) | 89.43 (0.63) | 97.33 (0.06) |
| | DropMessage | 94.03 (0.29) | 90.58 (0.64) | **97.75 (0.04)** |
| | M-Mixup | 93.47 (0.34) | 90.09 (0.92) | 96.22 (0.15) |
| | iGraphMix (*ours*) | **94.88 (0.40)** | **93.59 (0.45)** | 97.62 (0.05) |

the second-best performance on PubMed, the average performance ranking of our method was 1.33, supporting the effectiveness of our method on these datasets. This result may suggest that iGraphMix is effective in improving performance not only on the node classification but also on link prediction, showing the advantages of our method to be proved in various graph tasks.

## F    FURTHER DISCUSSIONS

Despite the effectiveness of our work, iGraphMix may have two limitations. First, the proposed method may not be as effective for certain types of graphs where the underlying assumption of our method may not hold. This limitation suggests designing an advanced and adaptive edge sampling method for tailoring it to each graph. Addressing it is a promising direction for future research, which could further optimize the applicability of iGraphMix across various graphs. Second, our theoretical analysis primarily focused on the two-layer GCN may not fully capture more complex and real-world scenarios. We believe that future works on extending multi-layer GCN could handle this limitation.

