# OpenReview forum: "iGraphMix: Input Graph Mixup Method for Node Classification"
_ICLR.cc/2024/Conference — ICLR 2024 poster_

### Official Review · Reviewer_vL8H · 2023-10-26

**Soundness:** 3 good
**Presentation:** 3 good
**Contribution:** 3 good
**Rating:** 6
**Confidence:** 4

**Summary:**

The paper proposes iGraphMix, a novel Mixup method tailored for node classification in graph neural networks (GNNs), which generates virtual nodes and edges by interpolating input features, labels, and neighboring nodes. iGraphMix addresses the irregularity and alignment issues associated with applying Input Mixup to node classification, and the paper provides theoretical proof and experimental results demonstrating its effectiveness in improving GNN performance and reducing overfitting.

**Strengths:**

* The paper provides theoretical proof and experimental validation of the effectiveness of iGraphMix in reducing the generalization gap and improving GNN performance

**Weaknesses:**

* The experimental validation of iGraphMix is mentioned, but it would be helpful to have more details on the datasets used, the specific GNN models employed, and the performance metrics used for evaluation.

* It would be beneficial to have experimental analysis about the computational cost and speed of the proposed method compared with the state-of-the-art approaches.

**Questions:**

* The paper mentions that iGraphMix can be combined with other augmentation methods. Can you provide examples or insights into how this combination can be done and what benefits it can bring to GNN performance?

---

> ### Author Response · Authors · 2023-11-19
> **Response to the reviewer vL8H (#1)**
>
> We appreciate the reviewer for recognizing our theoretical findings and experiments. Also, we would like to appreciate the feedback on the details of experiments and additional experiments, which help us improve the manuscript. Below we provide point-by-point responses.
>
> * * *
>
> ### **[W1]** (Details of experiments)
>
> Thanks for the reviewer’s suggestion. We will contain the details of the performance metric in Appendix C in the revised manuscript. We have included the basic information of the datasets and GNNs in Appendix C.1 and C.2, respectively.
>
> We used the micro-F1 score to compare the performance of the proposed method and other augmentation methods. Our choice of the performance metric is because the micro-F1 score, encompassing both precision and recall, offers a balanced perspective on model performance. This is particularly relevant in our context, where the understanding of true versus predicted class distributions is crucial.

---

> ### Author Response · Authors · 2023-11-19
> **Response to the reviewer vL8H (#2)**
>
> ***
> ### **[W2]** (Computational cost and speed)
>
> From Big O complexity analysis and the experiment, we indicated that our method does not exacerbate the computational speed. We will include the following result in the revised manuscript. Here is an in-depth analysis.
>
> We calculated the time complexity with the big O notation of all methods used in our paper. Let $\mathcal{V}$ be the number of nodes, $|\mathcal{E}|$ be the number of total edges, $|\mathcal{E}_{\mathrm{tr}}|$ be the number of edges connected to training nodes, $d$ be the hidden dimension of the GNN layer, and $K$ be the number of layers.
>
> Then, we can get the time complexity as shown in the below table. The time complexity of GCN was represented in the original GCN paper [1]. Since there are dropping edges or nodes in DropEdge and DropNode, the complexity for data preparation of DropEdge and DropNode is $\mathcal{O}(|\mathcal{E}|)$ and $\mathcal{O}(|\mathcal{E}|+\|\mathcal{V}\|)$, respectively. Also, the complexity for graph convolution is  $\mathcal{O}(p|\mathcal{E}|d^K)$, where $p \in [0, 1]$ is a dropping probability. DropMessage considers all edges and nodes in all layers, their complexity is the same as the standard GCN. For M-Mixup, the complexities for data preparation and graph convolution are $\mathcal{O}(\|\mathcal{E}\|+2\|\mathcal{V}\|)$ and  $\mathcal{O}(2|\mathcal{E}|d^K)$, respectively. Lastly, in the case of iGraphMix, the complexities for data preparation is $\mathcal{O}(\|\mathcal{E}\|+2\|\mathcal{V}\|)$ and that for graph convolution is $\mathcal{O}(\|\mathcal{E}\|d^K)$.
>
> | Operation           | None                              | DropEdge                           | DropNode                                       | DropMessage                        | M-Mixup                                         | iGraphMix (ours)                                |
> |---------------------|-----------------------------------|------------------------------------|------------------------------------------------|------------------------------------|-------------------------------------------------|-------------------------------------------------|
> | 1) Data Preparation | $\mathcal{O}(1)$                  | $\mathcal{O}(\|\mathcal{E}\|)$     | $\mathcal{O}(\|\mathcal{E}\|+\|\mathcal{V}\|)$ | $\mathcal{O}(1)$                   | $\mathcal{O}(\|\mathcal{E}\|+2\|\mathcal{V}\|)$ | $\mathcal{O}(\|\mathcal{E}\|+2\|\mathcal{V}\|)$ |
> | 2) Graph Convolution | $\mathcal{O}(\|\mathcal{E}\|d^K)$ | $\mathcal{O}(p\|\mathcal{E}\|d^K)$ | $\mathcal{O}(p\|\mathcal{E}\|d^K)$             | $\mathcal{O}(\|\mathcal{E}\|d^K)$ |  $\mathcal{O}(2\|\mathcal{E}\|d^K)$             | $\mathcal{O}(\|\mathcal{E}\|d^K)$               |
> |                     |                                   |                                    |                                                |                                    |                                                 |                                                 |
>
> From this analysis, we found that our method would not need lots of computational complexity, especially compared to M-Mixup. Although some methods require less computational time, our method leads to better performance when training GNNs, highlighting the necessity of the proposed method.
>
> The below table shows the computational time of data preparation and graph convolution on Cora and ogbn-arxiv datasets. Due to the time constraint, only the computational time for Cora and ogbn-arxiv was represented in this rebuttal instead of covering all the data. The below table shows the mean computational time (second) per epoch. We could find that the real computational time is consistent with our time complexity calculation. We will report these results in the appendix section of the revised manuscript.
>
> | Dataset    | Operation           | Data Augmentation |           |           |         |           ||
> |------------|---------------------|------------|-----------|-----------|---------|-----------|-----------|
> |            |                     | none       | DropEdge | DropNode |DropMessage|M-Mixup | iGraphMix |
> | Cora       | 1) Data preparation | **0.83e-3**    | 1.13e-3   | 1.33e-3   |**0.83e-3**| 1.96e-3 |  3.72e-3  |
> |            | 2) Graph Convolution         | 1.12e-2    | **1.10e-2**   | 1.10e-2   |1.11e-2| 1.73e-2 |  1.57e-2  |
> | ogbn-arxiv | 1) Data preparation | **1.25e-3**    | 1.34e-3   | 1.39e-3   | **1.25e-3** | 2.61e-2 |  2.56e-2  |
> |            | 2) Graph Convolution         | 3.32e-2    | **1.76e-2**   | 1.81e-2   | 3.30e-2 | 8.33e-2 |  3.37e-2  |

---

> ### Author Response · Authors · 2023-11-19
> **Response to the reviewer vL8H (#3)**
>
> ***
> ### **[Q1]** (Unifying iGraphMix with other augmentation methods)
>
> iGraphMix demonstrates its versatility by integrating with other augmentation methods applied to graph data, similar to MixMatch [2] which proposed the effective combination of augmentation methods and Mixup. First, we select an augmentation method and apply it to the original graph data. For instance, consider the application of DropNode that masks random node features. DropNode generates an augmented graph with some nodes removed from the original graph. Second, iGraphMix is applied to the augmented graph. In this step, with DropNode, iGraphMix performs interpolation on the remaining nodes and edges after DropNode. These two steps enable more diverse graph augmentation.
>
> The reason for the better result is due to the generation of more varied samples and the mitigation of distorted interrelationships between inputs and labels. By taking these advantages, the efficacy of graph augmentation can be enhanced, supported by our experiment. Furthermore, based on the results, exploring diverse sample generation for iGraphMix presents a promising research direction.
> ***
> [1] Kipf et al., Semi-supervised Classification with Graph Convolutional Networks, ICLR 2017
>
> [2] Berthelot et al., MixMatch: A Holistic Approach to Semi-Supervised Learning., NeurIPS, 2019.

---

> ### Author Response · Authors · 2023-11-22
> **Reminder Comment**
>
> Dear the reviewer vL8H,
>
>
> We would like to appreciate your time and thank you for your valuable feedback. As the deadline for the Author/Reviewer discussion is approaching, we kindly ask if you could let us know whether our responses have addressed your concerns. This will significantly help us improve our work. If you require further clarification, please do not hesitate to reach out to us.
>
>
> Sincerely yours,
>
> Authors of the paper 2291.

---

### Official Review · Reviewer_TxAH · 2023-10-27

**Soundness:** 2 fair
**Presentation:** 3 good
**Contribution:** 2 fair
**Rating:** 5
**Confidence:** 4

**Summary:**

The authors propose a new input mixup method for node classification problems. The proposed method, known as iGraphMix, generates virtual nodes by interpolating input features. The edges of these virtual nodes are generated by sampling neighboring nodes. The authors provide theoretical analysis to show that iGraphMix leads to better generalization performance compared to that without augmentation.

**Strengths:**

S1. The proposed method is easy to understand.

S2. The authors conduct extensive experiments to show that their proposed method outperforms multiple baselines.

S3. The authors provide a theoretical analysis of the generalization gap.

**Weaknesses:**

W1. The improvement of iGraphMix is marginal. Overall, the improvement beyond the second-best method is always less than 1%. I suggest the authors conduct experiments on more challenging datasets to make the result more convincing.

W2. How do the authors compute the generalization gap in Sec. 6.2? Why the test loss of iGraphMix is much higher than the "no augmentation"?

W3. In Appendix B, how can this $AX=AX'=A\tilde{X}$ holds? It would be much better if the authors could provide a rough proof idea before presenting all the details.

W4. The baselines compared are all very simple methods. There are more advanced graph data augmentation methods to compare with, such as [1].

W5. There are many existing graph mixup methods for graph classification tasks. It would be nice to add a discussion to better place this work in the literature.

[1] Kong, Kezhi, et al. "Robust optimization as data augmentation for large-scale graphs." Proceedings of the IEEE/CVF Conference on Computer Vision and Pattern Recognition. 2022.

**Questions:**

Q1. I don't understand why iGraphMix is versatile with other augmentation methods. Can authors provide more explanations?

Q2. Why do authors only use the Micro-F1 score as the only metric? Accuracy is a more common choice.

Q3. Does iGraphMix train GNNs using all virtual nodes, like how it is done in Mixup? In other words, no original nodes are used during training.

---

> ### Author Response · Authors · 2023-11-19
> **Response to the reviewer TxAH (#1)**
>
> We appreciate the reviewer for concerning our writing, experiments, and theoretical analysis. Also, we would like to appreciate the feedback on experiments and presentation details, which help us improve the manuscript. Below we provide point-by-point responses.
>
> * * *
> ### **[W1]** (Performance of iGraphMix and experiments on challenging datasets)
>
> #### 1) Performance of iGraphMix.
> While our method shows a small performance improvement compared to the second-best, iGraphMix can be synergized with other augmentation methods. The evidence is shown in Table 2; iGraphMix improves the performance with other augmentation methods by 1.86% on average. Our results show that our method can be easily unified with other augmentation methods to achieve additional performance improvements. Additionally, since this work can be a foundation work for future Input-Mixup research on graph-structured data, we can expect further performance improvement based on our method in future work.
>
> #### 2) Challenging Datasets
>
>
> | Datasets           | WikiCS           | Coauthor-CS |
> |--------------------|------------------|---------|
> | None               | 78.72 (0.18)     | 93.87 (0.02)  |
> | DropEdge           | 79.19 (0.13)     | 93.93 (0.06) |
> | DropNode           | 78.79 (0.34)     | 93.92 (0.05) |
> | DropMessage        | 79.79 (0.33)     | 93.26 (0.05) |
> | M-Mixup            | 79.76 (0.31)     | 94.08 (0.06) |
> | iGraphMix (_ours_) | **80.17 (0.11)** | **94.17 (0.06)** |
>
> The above table shows the micro-F1 score, which is equal to accuracy in multi-class case, for all augmentation methods on the two challenging datasets, WikiCS [1] and Coauthor-CS [2]. We found that iGraphMix outperformed other baselines on these datasets as well, suggesting that our method can be effective even for the challenging datasets. We also found that the average performance improvement of our method over the second-best method is 0.29% on average, while that of the second-best method over the third-best method is 0.10% on average. It indicates that our method may be better at improving performance than other augmentation methods.
>
> Furthermore, we have previously reported our experiment on large-scale graphs, Reddit and ogbn-product, in Appendix E.1. Since these graphs are significantly large compared to the transductive setting, we conducted inductive training and classification for these datasets. The results for these datasets are represented in Table 5. We found that iGraphMix shows better or comparable performance compared to other augmentation methods regardless of sub-graph sampling methods, backbone architectures, and datasets. It may imply that our method could be effectively applied to large-scale graphs.
> ***
> ### **[W2]** (Generalization gap of iGraphMix)
>
> We briefly explain how to calculate the generalization gap and why the test loss can be higher in iGraphMix compared to the standard training.
>
> #### 1) Calculation of the generalization gap.
>
> The generalization gap is calculated by the difference between the test loss and train loss as in Remark 3.1. The test loss and the train loss are defined as the cross-entropy loss of all test nodes and train nodes in the original graph (the reviewer can find them in Eqs. (2) and (3)).
>
> #### 2) Why can the test loss be higher in iGraphMix?
>
> The higher test loss in iGraphMix might be due to its smoothing effect of softmax output logits. This effect is because iGraphMix generates smoothed labels via label mixup, leading to smoothed output logits for training data. Given the common assumptions that training and test datasets are similarly distributed and that the training data is sufficiently large for generalization, the smoothing effect extends from the training dataset to the test dataset. Thus, we can find smoothed output logits for test data. While the smoothing effect can increase the test loss, it does not affect the maximum logit values, so there is a weak correlation between the test loss and the micro-F1 score (accuracy). Moreover, our theoretical and empirical findings demonstrate that iGraphMix narrows the generalization gap between training and test losses, indicating the proposed method's strong generalization capability​​.

---

> ### Author Response · Authors · 2023-11-19
> **Response to the reviewer TxAH (#2)**
>
> ***
> ### **[W3]** (Details about Appendix B.)
>
> #### 1) Clarification of $AX=AX’=A\tilde{X}$.
>
> We bring the matrix-level notation down to the node level to clarify what it means. Before explaining why the equation holds, we note that $A$ has no diagonal components. It means that there is no self-loop of all nodes. The equation is equivalent to $AX_v=AX'_v=A\tilde{X}_v \forall v$ in all labeled nodes. First, $AX_v=AX’_v$ is satisfied because nodes in the graph are permutation invariant. Second, we get $AX_v=A\tilde{X}_v$ because unlabeled nodes are not interpolated. These are valid for all labeled nodes, so we can combine them into the matrix notation.
>
> #### 2) Proof sketch
>
> The goal of our proof is to approximate the loss function of iGraphMix via second-order Taylor Approximation to understand the regularization effect (Lemma 5.2.). Here, $AX_v=AX’_v=A\tilde{X}_v$ is a crucial precondition for deriving Lemma B.1. and B.2. These lemmas are necessary to make zero of a first-order term of the approximated loss, leading us to the desired regularization term. Consequently, we get the regularization term and define the weight space of GCN trained with iGraphMix as a related dual form of the regularization term as illustrated in equation (7). Considering the weight space into Transductive Rademacher Complexity (TRC), we could get Theorem 5.3., which reveals the upper bound of the generalization gap of iGraphMix. Subsequently, with appropriate beta distribution parameters, we can make the lower generalization upper bound, showing the effectiveness of our method.
>
> ***
> ### **[W4]** (Simple baselines)
>
> We understand the importance of comparing our method with a wide range of baselines, including Kong et al [3]. However, the most important factor we considered in selecting a baseline was to ensure a direct and fair comparison of data augmentation methods under similar experimental conditions. The inclusion of methods with additional training techniques (e.g., min-max optimization, and robust optimization) violates our consideration. Our concern was that integrating those methods without their unique training techniques may distort their original intention, and it could potentially make a skewed comparison. However, since including previous work that the reviewer mentioned [3] is crucial, we will briefly summarize and highlight this work’s importance in the related work section of the revised manuscript.
>
> ***
> ### **[W5]** (Related works on other tasks)
>
>
> While numerous mix-up methods [4,5,6,7,8] have been proposed for graph classification, our related work section does not include them for the following reason. They primarily focus on node matching across different graphs, which is out-of-scope. Also, as their application to node classification is limited, we exclude them from our related work.
>
> ***
> ### **[Q1]** (Versatility of iGraphMix)
>
> iGraphMix demonstrates its versatility by integrating with other augmentation methods applied to graph data, similar to MixMatch [9] which proposed the effective combination of augmentation methods and Mixup. This integration is executed as the following process.
>
> First, we select an augmentation method and apply it to the original graph data. For instance, consider the application of DropNode that masks random node features. DropNode generates an augmented graph with some nodes removed from the original graph.
>
> Second, iGraphMix is applied to the augmented graph. In this step, iGraphMix performs interpolation on the remaining nodes and edges after the DropNode. These two steps enable more diverse graph augmentation.
> ***
> ### **[Q2]** (Evaluation metric)
>
> To address the concern regarding our choice of the micro-F1 score over accuracy, we would like to explain the reasons as follows. First, in multi-class classification tasks, such as our paper’s task, the micro-F1 score and accuracy are equivalent. This is because each data point is unequivocally assigned to one class, and it is described in detail in the paper [10]. Second, although their values are the same, we would like to denote it as a micro-F1 score rather than an accuracy. The micro-F1 score, encompassing both precision and recall, offers a balanced perspective on model performance. This is particularly relevant in our context, where the understanding of true versus predicted class distributions is crucial. Last, there are many references [10, 11] to represent the performance as micro-F1 score in the node classification. Therefore, we represent the performance using the micro-F1 score in our paper.

---

> ### Author Response · Authors · 2023-11-19
> **Response to the reviewer TxAH (#3)**
>
> ***
> ### **[Q3]** (Details about training nodes for iGraphMix)
>
> Similar to other mixup studies, the GNNs trained with iGraphMix use only virtual nodes.  The original nodes are used indirectly for training as they are mixed in with the virtual nodes.
>
> ***
>
> [1] Mernyei et al., Wiki-CS: A Wikipedia-Based Benchmark for Graph Neural Networks, ICML, 2021
>
> [2] Shchur et al., Pitfalls of Graph Neural Network Evaluation, arXiv preprint arXiv:1811.05868, 2018.
>
> [3] Kong, Kezhi, et al. "Robust optimization as data augmentation for large-scale graphs.", CVPR, 2022.
>
> [4] Han et al., G-Mixup: Graph Data Augmentation for Graph Classification, ICML, 2022.
>
> [5] Park et al., Graph Transplant: Node Saliency-Guided Graph Mixup with Local Structure Preservation, AAAI, 2022.
>
> [6] Yoo et al., Model-Augnostic Augmentation for Accurate Graph Classification, WWW, 2022.
>
> [7] Guo \& Mao, ifMixup: Interpolating Graph Pair to Regularize Graph Classification, AAAI, 2023.
>
> [8] Ling et al., Graph Mixup with Soft Alignments, ICML, 2023.
>
> [9] Berthelot et al., MixMatch: A Holistic Approach to Semi-Supervised Learning., NeurIPS, 2019.
>
> [10] Zhao et al., Data Augmentation for Graph Neural Networks, AAAI 2020.
>
> [11] Wang et al., Mixup for Node and Graph Classification, WWW, 2021.

---

> ### Author Response · Authors · 2023-11-22
> **Reminder Comment**
>
> Dear the reviewer TxAH,
>
>
> We would like to appreciate your time and thank you for your valuable feedback. As the deadline for the Author/Reviewer discussion is approaching, we kindly ask if you could let us know whether our responses have addressed your concerns. This will significantly help us improve our work. If you require further clarification, please do not hesitate to reach out to us.
>
>
> Sincerely yours,
>
> Authors.

---

> > ### Comment · Area_Chair_HffJ · 2023-12-03
> >
> > Dear reviewer TxAH,
> >
> > For reviewer TxAH, your score is 3 which differs from others score much. So please read the authors' response and other reviewers' comments to check whether you would like to change your score. Thanks.
> >
> > Bests,
> > AC

---

### Official Review · Reviewer_jPtV · 2023-10-28

**Soundness:** 3 good
**Presentation:** 2 fair
**Contribution:** 3 good
**Rating:** 6
**Confidence:** 4

**Summary:**

This paper proposes a node-level graph mixup method named iGraphMix to improve the model generation ability. To handle the irregularity and alignment issue for graph mixup, this paper proposes to generate virtual nodes and edges by interpolating features and labels, and attaching sampled neighborhoods. Theoretical analysis shows that iGraphMixup can be regarded as a regularization on the weight space to help improve the generalization. Experiments on real world datasets validate the effectiveness of the proposed method on node classification.

**Strengths:**

-	A novel method is proposed to mixup graphs at the input level.
-	Theoretical analysis is provided to understand the effect of improving the model generalization.
-	Extensive experiments are provided to evaluate the method empirically.

**Weaknesses:**

-	Baseline methods are quite limited and evaluation on robustness is highly recommended. See details in the question part.
-	Presentation could be further improved.

**Questions:**

-	How will the proposed method enhance the model robustness? Robustness w.r.t label/feature/structure noises is usually evaluated for mixup methods [1,2], and it is highly recommended to include these experiments in the paper.
-	More baselines are needed. Currently, only M-mixup is a graph mixup for node classification, while other augmentation methods (e.g., [4]) are not included.
-	Eq.(4): How can A,X and its permuted counterpart A’,X’ be directly added as they are not well-aligned? Is the masking matrix M a symmetric matrix?
-	Writting:
  - Eq.(6), notations $\tilde{Z}_{v,v’}$, $\tilde{Y}_{v,v’}$ is quite misleading, as subscripts are used to denote columns and rows in the paper.
  - Line below eq.(1): matrix->matrices.

Reference

[1] Han, Xiaotian, et al. "G-mixup: Graph data augmentation for graph classification." International Conference on Machine Learning. PMLR, 2022.

[2] Ling, Hongyi, et al. "Graph Mixup with Soft Alignments." arXiv preprint arXiv:2306.06788 (2023).

[3] Pascal Esser, Leena Chennuru Vankadara, and Debarghya Ghoshdastidar. Learning theory can (sometimes) explain generalisation in graph neural networks. Advances in Neural Information Processing Systems, 34:27043–27056, 2021.

[4] Verma, Vikas, et al. "Graphmix: Improved training of gnns for semi-supervised learning." Proceedings of the AAAI conference on artificial intelligence. Vol. 35. No. 11. 2021.

[5] Wu, Lirong, et al. "Graphmixup: Improving class-imbalanced node classification by reinforcement mixup and self-supervised context prediction." Joint European Conference on Machine Learning and Knowledge Discovery in Databases. Cham: Springer Nature Switzerland, 2022.

---

> ### Author Response · Authors · 2023-11-19
> **Response to the reviewer jPtV (#1)**
>
> We appreciate the reviewer for the recognition of our novelty and analysis. Also, we would like to appreciate the guidance provided on experiments and writing quality, which help us improve the manuscript. Below we provide point-by-point responses. We respond to some weaknesses and questions simultaneously because of their similarity.
>
> * * *
>
> ### **[W1 & Q1]** (Robustness)
>
> To evaluate the robustness, we conducted two additional experiments on Cora, CiteSeer, and PubMed. For all experiments, we used GCN as the backbone model. We will include these results in the Appendix of our revised manuscript. We hope these experiments may resolve the reviewer’s concern.
>
> #### 1) Feature noise
>
> We evaluated the robustness when the feature noise is introduced in the graph. The feature noise is provided by adding standard Gaussian noise to input features with a probability of noise level. The below table shows how robust our method is for the feature noise. We indicated that our method outperforms the standard training in all noise levels.
>
>
> | Dataset   | Cora         |              |              | CiteSeer     |              |              | PubMed       |              |              |
> |-----------|--------------|--------------|--------------|--------------|--------------|--------------|--------------|--------------|--------------|
> | Noise level (%) | 10%          | 20%          | 40%          | 10%          | 20%          | 40%          | 10%          | 20%          | 40%          |
> | None      | 66.90 (1.54) | 66.89 (1.72) | 66.12 (1.73) | 78.10 (1.47) | 77.76 (1.54) | 77.92 (1.58) | 75.46 (1.67) | 74.62 (1.96) | 73.36 (2.49) |
> | iGraphMix | 68.48 (1.70) | 68.54 (1.58) | 68.30 (2.08) | 79.00 (2.41) | 78.64 (2.29) | 78.76 (2.94) | 75.78 (1.90) | 75.80 (2.03) | 74.88 (2.37) |
>
> #### 2) Structural Noise
>
> We assessed the robustness when the graph contains structural noise. The structural noise is given by randomly adding and removing edges with a probability of noise level. The below table shows the results of how the proposed method is robust to the structural noise. We found that when structural noise is introduced, GCN trained with iGraphMix outperforms GCN trained without iGraphMix (None). In addition, as the noise level increases from 10% to 40%, our method achieves more performance improvement compared to the standard training (None).
>
> | Dataset   | Cora         |              |              | CiteSeer     |              |              | PubMed       |              |              |
> |-----------|--------------|--------------|--------------|--------------|--------------|--------------|--------------|--------------|--------------|
> | Noise level (%) | 10%          | 20%          | 40%          | 10%          | 20%          | 40%          | 10%          | 20%          | 40%          |
> | None      | 67.62 (1.02) | 61.88 (1.10) | 54.90 (1.97) | 78.62 (0.56) | 73.14 (0.37) | 60.96 (0.84) | 73.92 (0.37) | 71.88 (0.53) | 60.20 (0.58) |
> | iGraphMix | 68.94 (1.11) | 63.22 (0.38) | 55.82 (0.46) | 79.14 (0.47) | 74.06 (0.31) | 61.78 (0.82) | 74.54 (0.10) | 72.56 (0.54) | 61.22 (0.61) |
>
> #### 3) Label Noise
>
> We assessed the robustness when the graph contains label noise.  We randomly permuted training labels with a probability of noise level for this experiment.  We trained GCN with the noisy training labels and evaluated GCN's performance. The below table shows how robust our method for noise labels is. We indicated that our method outperforms the standard training in all noise levels.
>
> | Dataset   | Cora         |              |              | CiteSeer     |              |              | PubMed       |              |              |
> |-----------|--------------|--------------|--------------|--------------|--------------|--------------|--------------|--------------|--------------|
> | Noise level | 10%          | 20%          | 40%          | 10%          | 20%          | 40%          | 10%          | 20%          | 40%          |
> | None      | 79.10 (0.74) | 76.00 (1.07) | 67.60 (2.72) | 68.24 (2.40) | 64.14 (2.04) | 57.72 (4.65) | 77.32 (1.36) | 75.76 (2.52) | 68.64 (3.15) |
> | iGraphMix | 80.02 (0.86) | 76.42 (1.54) | 69.08 (2.11) | 69.08 (1.97) | 65.92 (2.30) | 58.62 (3.48) | 77.66 (1.55) | 76.00 (2.64) | 69.04 (3.59) |

---

> ### Author Response · Authors · 2023-11-19
> **Response to the reviewer jPtV (#2)**
>
> ***
> ### **[W1 & Q2]** (Baseline methods)
>
> We understand the importance of comparing our method with a wide range of baselines, including Verma et al [1]. However, the most important factor we considered in selecting a baseline was to ensure a direct and fair comparison of data augmentation methods under similar experimental conditions. The inclusion of methods with additional training techniques (e.g., co-training, using the original graph for training, and auxiliary losses and models for manifold mixup) violates our consideration. Our concern was that integrating those methods without their unique training techniques may distort their original intention, and it could potentially make a skewed comparison. However, since including previous work that the reviewer mentioned [1] is crucial, we briefly summarized and highlighted this work’s importance in the related work section.
>
> ***
> ### **[Q3]** (Details about Eq. (4))
>
> We clarify that the direct addition of $M_\lambda \cdot A$, $X$ and its permuted counterpart $M_{1-\lambda} \cdot A’$, $X’$ is feasible since the size of each matrix remains the same even if the rows and columns are permuted. Specifically, when we apply mixup during the training, there is an alignment issue between neighboring nodes, but we are simply addressing the alignment issue by concatenating and constructing new edge sets in the new virtual graph through a simple direct addition. In addition, masking matrices do not need to be symmetric.
> ***
> ### **[W2 & Q4]** (Writing quality)
>
> We appreciate the reviewer’s suggestion. We will revise $\tilde{Y}{v,v’}$ to $\tilde{Y}{v}^{v’}$ for resolving some misleading problem. Moreover, we will double-check and fix typos, such as the ‘matrix’ line below Eq.(1), in the revised manuscript. Again, thanks for the reviewer’s advice.
>
> ***
> [1] Verma et al., Improved Training of GNNs for Semi-Supervised Learning, AAAI, 2021.

---

> ### Author Response · Authors · 2023-11-22
> **Reminder Comment**
>
> Dear the reviewer jPtV,
>
>
> We would like to appreciate your time and thank you for your valuable feedback. As the deadline for the Author/Reviewer discussion is approaching, we kindly ask if you could let us know whether our responses have addressed your concerns. This will significantly help us improve our work. If you require further clarification, please do not hesitate to reach out to us.
>
>
> Sincerely yours,
>
> Authors.

---

> > ### Comment · Reviewer_jPtV · 2023-11-22
> >
> > Thank you for your clarifications and additional experiments, which should be included in the future version. For the robustness experiments, please also include robustness results of other (at least some representative) baselines. Besides, I would still encourage the author to include more baselines to help better empirically evaluate the proposed method.
> >
> > Overall, I think this is a theorectically sounded work but more empirical evaluation is highly recommended in the future version. Given the current version, I'm not able to give it a clear accept (8) and I'd like to keep my current rating (6), but I champion this paper to be accepted.

---

> ### Author Response · Authors · 2023-11-23
>
> Dear the reviewer jPtV,
>
> Thank you for the insightful feedback. We are committed to incorporating your suggestions into the revised manuscript. Specifically, in addition to the original response (https://openreview.net/forum?id=a2ljjXeDcE&noteId=Tr5QDvUSTT), we will include additional robustness results for other baselines. Also, we will expand our empirical evaluation with more baselines such as ifMixup [1], which was originally proposed for graph classification but can be easily adapted to node classification. The below table shows the results of ifMixup on CiteSeer and Cora datasets. Although we have conducted the experiment on the two datasets due to time constraints, we plan to extend this to the additional datasets in the revised manuscript. This additional analysis will be integrated into the revised manuscript. We acknowledge your suggestion to include GraphMix [2] as a baseline. However, we omit it in our manuscript to avoid distorting the original paper's intention, as GraphMix involves additional training techniques. We acknowledge the approaching deadline for the Author/Reviewer discussion and will make these enhancements as promptly as possible to strengthen our work.
>
> We are grateful for your recognition of the theoretical soundness of our paper and understand your stance on the current rating. Your support and constructive guidance have been invaluable in refining our manuscript for its future iteration.
>
> | Augmentation Methods | CiteSeer         | Cora             |
> |--------------------------|------------------|------------------|
> | None                     | 72.05 (0.56)     | 82.65 (0.55)     |
> | ifMixup [1]              | 71.89 (1.04)     | 81.27 (0.75)     |
> | iGraphMix (proposed) | **73.67 (0.61)** | **83.78 (0.42)** |
>
>
> Sincerely yours,
>
> Authors.
>
> ***
>
> [1] Guo et al., Interpolating graph pair to regularize graph classification, AAAI, 2023.
>
> [2] Verma et al., Improved Training of GNNs for Semi-Supervised Learning, AAAI, 2021.

---

### Official Review · Reviewer_mUwv · 2023-11-01

**Soundness:** 3 good
**Presentation:** 3 good
**Contribution:** 3 good
**Rating:** 8
**Confidence:** 5

**Summary:**

This paper presents a new method called iGraphMix for node classification in graph neural networks. The method addresses the challenges of irregularity and alignment in generating virtual nodes and edges for GNNs training. iGraphMix generates virtual graphs that serve as inputs for GNNs training, leading to better generalization performance compared to training without augmentation. The authors evaluate iGraphMix on several benchmark datasets and show that it outperforms existing state-of-the-art methods. The contributions of this paper include a novel approach to graph augmentation, a comprehensive evaluation of the proposed method, and insights into the effectiveness of virtual graph generation for GNNs training.

**Strengths:**

This paper presents a novel method, iGraphMix, for addressing the challenges of irregularity and alignment in generating virtual nodes and edges for graph neural networks. The method is well-motivated and builds on existing work in Input Mixup for other domains. The authors provide a clear and comprehensive description of the method, including theoretical analysis and experimental validation of its effectiveness. The evaluation is thorough and includes comparisons to existing state-of-the-art methods on several benchmark datasets. The results show that iGraphMix outperforms existing methods in terms of micro-F1 score, demonstrating the significance of the proposed approach.

Overall, the paper is well-written and easy to follow, with clear explanations of the technical details and experimental setup. The authors provide a detailed discussion of related work and highlight the contributions of their method. The theoretical analysis is insightful and provides a deeper understanding of the effectiveness of iGraphMix. The experimental results are convincing and demonstrate the superiority of iGraphMix over existing methods.

In terms of originality, iGraphMix is a novel approach to graph augmentation that addresses the challenges of irregularity and alignment in generating virtual nodes and edges for GNNs training. The method builds on existing work in Input Mixup for other domains but is specifically designed for node classification in the graph domain. The authors provide a clear motivation for the method and demonstrate its effectiveness through theoretical analysis and experimental validation.

In terms of quality, the paper is well-written and well-organized, with clear explanations of the technical details and experimental setup. The authors provide a thorough evaluation of the proposed method, including comparisons to existing state-of-the-art methods on several benchmark datasets. The results are convincing and demonstrate the superiority of iGraphMix over existing methods.

In terms of clarity, the paper is easy to follow, with clear explanations of the technical details and experimental setup. The authors provide a detailed discussion of related work and highlight the contributions of their method. The theoretical analysis is insightful and provides a deeper understanding of the effectiveness of iGraphMix.

In terms of significance, the paper presents a novel approach to graph augmentation that addresses the challenges of irregularity and alignment in generating virtual nodes and edges for GNNs training. The method is well-motivated and builds on existing work in Input Mixup for other domains. The authors provide a clear motivation for the method and demonstrate its effectiveness through theoretical analysis and experimental validation. The results show that iGraphMix outperforms existing methods in terms of micro-F1 score, demonstrating the significance of the proposed approach.

**Weaknesses:**

Overall, the paper is well-written and presents a novel approach to graph augmentation for node classification in GNNs. However, there are a few weaknesses that could be addressed to improve the paper:

1. Limited analysis of the impact of hyperparameters: The authors do not provide a detailed analysis of the impact of hyperparameters on the performance of iGraphMix. It would be useful to see how the performance of iGraphMix varies with different hyperparameters, such as the number of virtual nodes or the strength of the mixing coefficient.

2. Lack of ablation study: The authors do not provide an ablation study to analyze the contribution of each component of iGraphMix. It would be useful to see how the performance of iGraphMix varies when different components are removed or modified.

3. Limited discussion of limitations: The authors do not provide a detailed discussion of the limitations of iGraphMix. It would be useful to see a discussion of the scenarios where iGraphMix may not be effective or where other methods may be more appropriate.

4. Lack of analysis of computational complexity: The authors do not provide an analysis of the computational complexity of iGraphMix. It would be useful to see how the computational cost of iGraphMix compares to other graph augmentation methods and how it scales with the size of the graph.

Addressing these weaknesses would strengthen the paper and provide a more comprehensive evaluation of the proposed method.

**Questions:**

How sensitive is the performance of iGraphMix to the choice of hyperparameters, such as the number of virtual nodes or the strength of the mixing coefficient? Can the authors provide a detailed analysis of the impact of hyperparameters on the performance of iGraphMix?

Can the authors provide an ablation study to analyze the contribution of each component of iGraphMix? This would help to better understand the importance of each component and how the performance of iGraphMix varies when different components are removed or modified.

What are the limitations of iGraphMix? Can the authors provide a detailed discussion of the scenarios where iGraphMix may not be effective or where other methods may be more appropriate?

Can the authors provide an analysis of the computational complexity of iGraphMix? How does the computational cost of iGraphMix compare to other graph augmentation methods, and how does it scale with the size of the graph?

How does iGraphMix perform on larger and more complex graphs? Can the authors provide an analysis of the scalability of iGraphMix to larger graphs with more nodes and edges?

Can the authors provide a discussion of the potential applications of iGraphMix beyond node classification, such as link prediction or graph classification?

How does iGraphMix perform on graphs with different characteristics, such as sparsity or degree distribution? Can the authors provide an analysis of the robustness of iGraphMix to different graph properties?

Can the authors provide a discussion of the potential limitations of the theoretical analysis presented in the paper? How well does the theoretical analysis capture the behavior of iGraphMix in practice?

Can the authors provide a discussion of the potential ethical implications of using graph augmentation methods like iGraphMix? How can we ensure that these methods are used responsibly and do not perpetuate biases or inequalities in the data?

---

> ### Author Response · Authors · 2023-11-19
> **Response to reviewer mUwv (#1)**
>
> We would like to appreciate the reviewer providing constructive feedback and detailed suggestions, which help us improve the manuscript. Below, we provide point-by-point responses. Some weaknesses and questions are addressed simultaneously due to their similarity.
>
> * * *
>
> ### **[W1, Q1]** (Sensitivity analysis on iGraphMix)
>
> We have conducted the sensitivity analysis of $\beta$ in Figure 3 because it is the only hyperparameter for the proposed method. In addition to $\beta$, we have experimented with the number of layers $K$, a hyperparameter for GNNs, in Figure 4(a). We also have analyzed the number of labeled nodes per class $L$ in Figure 4(b) since the performance of GNNs is related to $L$.
>
> Figure 3 demonstrates a consistent reduction in the generalization gap according to the various beta. Notably, this gap becomes larger as beta increases, signifying more well-mixed graphs via iGraphMix. However, as beta decreases, the performance of iGraphMix approaches that of standard training, suggesting a lower bound where the benefits of mixup are less pronounced. This observation highlights the importance of a well-calibrated mixing coefficient to fully harness the potential of iGraphMix.
>
> ***
>
> ### **[W2, Q2]** (Ablation study on iGraphMix)
>
> | Datasets       | CiteSeer     | Cora         | PubMed       |
> |-----------------|--------------|--------------|--------------|
> | None            | 72.05 (0.56) | 82.65 (0.55) | 79.32 (0.15) |
> | iGraphMix (input+label) | 71.50 (0.47) | 81.26 (0.27) | 79.18 (0.53) |
> | iGraphMix (edge+label)  | 72.44 (0.81) | 82.72 (0.21) | 79.18 (0.33) |
> | iGraphMix (input+edge+label) (_proposed_)  | **73.67 (0.61)** | **83.78 (0.42)** | **79.93 (0.60)** |
>
> We conducted the ablation study on iGraphMix and the result is summarized in the above table. Specifically, we compared our method with 1) applying mixup only to input and label and 2) applying mixup only to edge and label on three transductive datasets for node classification. We observed that iGraphMix (input + edge+ label) consistently outperforms not only the standard training but also iGraphMix (input+label), and iGraphMix (edge+label). It may imply that interpolating all components of the graph is crucial, showing our method’s effectiveness. Furthermore, iGraphMix (input+label) consistently underperformed the standard training, and iGraphMix (edge+label) is better or comparable to the standard training. It may imply the importance of applying mixup on the edge.
>
> ***
>
> ### **[W3, Q3, Q8]** (Discussions on the limitations of this work)
>
> We appreciate the opportunity to address the importance of the discussion. We will include this discussion in the revised manuscript. Here is the potential limitation of our method and theoretical analysis.
>
> #### 1) The limitation of our method
>
> iGraphMix may not be as effective for certain types of graphs where the underlying assumption of our method may not hold. This limitation suggests us to design an advanced and adaptive edge sampling method for tailoring it to each graph. Addressing it is a promising direction for future research, which could further optimize the applicability of iGraphMix across various graphs.
>
> #### 2) The limitation of the theoretical analysis
>
> Our theoretical analysis primarily focused on the two-layer GCN may not fully capture more complex and real-world scenarios. We believe that future works on extending multi-layer GCN could handle this limitation.

---

> ### Author Response · Authors · 2023-11-19
> **Response to reviewer mUwv (#2)**
>
> ***
> ### **[W4, Q4]** (Time complexity)
>
> From Big O complexity analysis and the experiment, we indicated that our method does not exacerbate the computational speed. We will include the following result in the revised manuscript. Here is an in-depth analysis.
>
> We calculated the time complexity with the big O notation of all methods used in our paper. Let $\mathcal{V}$ be the number of nodes, $|\mathcal{E}|$ be the number of total edges, $|\mathcal{E}_{\mathrm{tr}}|$ be the number of edges connected to training nodes, $d$ be the hidden dimension of the GNN layer, and $K$ be the number of layers.
>
> Then, we can get the time complexity as shown in the below table. The time complexity of GCN was represented in the original GCN paper [1]. Since there are dropping edges or nodes in DropEdge and DropNode, the complexity for data preparation of DropEdge and DropNode is $\mathcal{O}(|\mathcal{E}|)$ and $\mathcal{O}(|\mathcal{E}|+\|\mathcal{V}\|)$, respectively. Also, the complexity of graph convolution is  $\mathcal{O}(p|\mathcal{E}|d^K)$, where $p \in [0, 1]$ is a dropping probability. DropMessage considers all edges and nodes in all layers, their complexity is the same as the standard GCN. For M-Mixup, the complexities for data preparation and graph convolution are $\mathcal{O}(\|\mathcal{E}\|+2\|\mathcal{V}\|)$ and  $\mathcal{O}(2|\mathcal{E}|d^K)$, respectively. Lastly, in the case of iGraphMix, the complexities for data preparation is $\mathcal{O}(\|\mathcal{E}\|+2\|\mathcal{V}\|)$ and that for graph convolution is $\mathcal{O}(\|\mathcal{E}\|d^K)$.
>
> | Operation           | None                              | DropEdge                           | DropNode                                       | DropMessage                        | M-Mixup                                         | iGraphMix (ours)                                |
> |---------------------|-----------------------------------|------------------------------------|------------------------------------------------|------------------------------------|-------------------------------------------------|-------------------------------------------------|
> | 1) Data Preparation | $\mathcal{O}(1)$                  | $\mathcal{O}(\|\mathcal{E}\|)$     | $\mathcal{O}(\|\mathcal{E}\|+\|\mathcal{V}\|)$ | $\mathcal{O}(1)$                   | $\mathcal{O}(\|\mathcal{E}\|+2\|\mathcal{V}\|)$ | $\mathcal{O}(\|\mathcal{E}\|+2\|\mathcal{V}\|)$ |
> | 2) Graph Convolution | $\mathcal{O}(\|\mathcal{E}\|d^K)$ | $\mathcal{O}(p\|\mathcal{E}\|d^K)$ | $\mathcal{O}(p\|\mathcal{E}\|d^K)$             | $\mathcal{O}(\|\mathcal{E}\|d^K)$ |  $\mathcal{O}(2\|\mathcal{E}\|d^K)$             | $\mathcal{O}(\|\mathcal{E}\|d^K)$               |
> |                     |                                   |                                    |                                                |                                    |                                                 |                                                 |
>
> From this analysis, we found that our method would not need lots of computational complexity, especially compared to M-Mixup. Although some methods require less computational time, our method leads to better performance when training GNNs, highlighting the necessity of the proposed method.
>
> The below table shows the computational time of data preparation and graph convolution on Cora and ogbn-arxiv datasets. Due to the time constraint, only the computational time for Cora and ogbn-arxiv was represented in this rebuttal instead of covering all the data. The below table shows the mean computational time (second) per epoch. We could find that the real computational time is consistent with our time complexity calculation. We will report these results in the appendix section of the revised manuscript.
>
> | Dataset    | Operation           | Data Augmentation |           |           |         |           | |
> |------------|---------------------|------------|-----------|-----------|---------|-----------|-----------|
> |            |                     | none       | DropEdge | DropNode |DropMessage|M-Mixup | iGraphMix |
> | Cora       | 1) Data preparation | **0.83e-3**    | 1.13e-3   | 1.33e-3   |**0.83e-3**| 1.96e-3 |  3.72e-3  |
> |            | 2) Graph Convolution         | 1.12e-2    | **1.10e-2**   | 1.10e-2   |1.11e-2| 1.73e-2 |  1.57e-2  |
> | ogbn-arxiv | 1) Data preparation | **1.25e-3**    | 1.34e-3   | 1.39e-3   | **1.25e-3** | 2.61e-2 |  2.56e-2  |
> |            | 2) Graph Convolution         | 3.32e-2    | **1.76e-2**   | 1.81e-2   | 3.30e-2 | 8.33e-2 |  3.37e-2  |

---

> ### Author Response · Authors · 2023-11-19
> **Response to reviewer mUwv (#3)**
>
> ***
> ### **[Q5]** (iGraphMix on large graphs)
>
> Due to the paper limitation, we reported our experiment on large-scale graphs, such as Reddit and ogbn-product, in Appendix E.1. Since these graphs are significantly large for considering the transductive setting, we conducted inductive training and classification on these datasets. It means that we sampled sub-graphs from whole graphs and trained GNNs via those graphs. The results for these datasets are represented in Table 5. We found that iGraphMix is better or comparable to other augmentation methods regardless of sub-graph sampling methods, backbone architectures, and datasets. It may imply that our method could be effectively applied to large-scale graphs.
> ***
> ### **[Q6]** (iGraphMix on other tasks)
>
>  In Appendix E.2., we have included a detailed experiment on link prediction. Our results are summarized in Table 6. It demonstrates that iGraphMix not only outperforms in node classification but also shows significant improvement in link prediction compared to other augmentation methods. It can be evidence of the extension of our method to other graph-based tasks.  We leave the application of iGraphmix to graph classification to future work as mixing two different graphs necessitates addressing the complexities of node matching across different graphs.
> ***
>
> ### **[Q7]** (iGraphMix on graphs with various characteristics)
>
> | Datasets     | Cora  | CiteSeer | PubMed | ogbn-arxiv | Flickr |
> |--------------|-------|----------|--------|------------|--------|
> | Sparsity (%) | 0.082 | 0.144    | 0.023  | 0.008      | 0.011  |
> | Avg. Degree  | 2.74  | 3.90     | 4.50   | 2.74       | 10.08  |
>
> In the table above, we have examined the sparsity and the average degree of all the graph datasets used in our experiment. We discovered that the datasets exhibit varying levels of sparsity and average degree. Upon correlating these statistics with our findings, we have confirmed that iGraphMix has the potential to substantially enhance GNNs’ performance.
> ***
>
> ### **[Q9]** (Ethical consideration)
>
> We realize that ensuring our method may not mitigate biases or inequalities. Our work initially focused on the technical aspects and did not explicitly address potential biases or inequalities that might arise. However, we think that there can be a potential future direction to address biases and inequalities in algorithms. For instance, similar to Fair Mixup [2], our method can be improved to mitigate potential harmful biases in node classification.
>
> ***
> [1] Kipf et al., Semi-supervised Classification with Graph Convolutional Networks, ICLR 2017
>
> [2] Chuang et al., Fair Mixup: Fairness via Interpolation, ICLR, 2021

---

> ### Author Response · Authors · 2023-11-22
> **Reminder Comment**
>
> Dear the reviewer mUwv,
>
> We would like to appreciate your time and thank you for your valuable feedback. As the deadline for the Author/Reviewer discussion is approaching, we kindly ask if you could let us know whether our responses have addressed your concerns. This will significantly help us improve our work. If you require further clarification, please do not hesitate to reach out to us.
>
> Sincerely yours,
>
> Authors.

---

### Official Review · Reviewer_A2TK · 2023-11-02

**Soundness:** 3 good
**Presentation:** 3 good
**Contribution:** 4 excellent
**Rating:** 6
**Confidence:** 3

**Summary:**

This paper proposed iGraphMix that addresses the irregularity and alignment issues of Input Mixup on node classification. Specifically, to address the two issues, iGraphMix does not only interpolate node features and labels but also aggregates the sampled neighboring nodes. Theoretical analysis of the generalization gap and related experiments on the real-world graphs showed that the proposed method is effective in regularizing GNNs by generating diverse virtual samples and preserving high usability and versatility.

**Strengths:**

1. The paper is well organized and theoretical.
2. The proposed method iGraphMix is simple but effective.

**Weaknesses:**

1. In Section 5 THEORETICAL ANALYSIS, the author mentioned that “Citeseer dataset contains only 1.71% connected edges of labeled nodes out of all edges”, but the data “1.71%” lacks of related references.
2. Considering iGraphMix that the essence of iGraphMix is to implement a mixed strategy for features, labels and adjacency matrix respectively, however, the experiment content lacks the ablation experiment for these three components. It would be better to add related ablation experiments to examine the effect of these three components.

**Questions:**

From the perspective of time complexity, how does the time cost of iGraphMix compare with other augmentation methods? Can you add a diagram to show it?

---

> ### Author Response · Authors · 2023-11-19
> **Response to the reviewer A2TK (#1)**
>
> We would like to appreciate the reviewer's concern for providing constructive feedback and detailed suggestions, which help us improve the manuscript. Below we provide point-by-point responses.
> ***
> ### **[W1]** (Edges between labeled Nodes)
>
> |                                            | Datasets |       |        |            |        |
> |--------------------------------------------|----------|-------|--------|------------|--------|
> | Statistics                                 | CiteSeer | Cora  | PubMed | ogbn-arxiv | Flickr |
> | Num. Edges                                 | 4552     | 5278  | 88648  | 1166243    | 899756 |
> | Num. Labeled Nodes                         | 120      | 140   | 60     | 9135       | 4534   |
> | Num. Edges Between Labeled Nodes | 81       | 21    | 0      | 3664       | 1952   |
> | Edges Between Labeled Nodes Ratio (%) | 1.78%    | 0.40% | 0.00%     | 0.32%      | 0.21%  |
>
> The above table describes the number of edges, labeled nodes, connected edges between labeled nodes, and the ratio of connected edges between labeled nodes for five transductive datasets that we used for the node classification experiment. The edges between labeled nodes ratio means the number of edges between labeled nodes divided by the total number of edges. Since there are few numbers of connected edges between labeled nodes, the assumption of no connection between labeled nodes seems reasonable for the theoretical analysis. We will include this additional information in Appendix C in the revised manuscript.

---

> ### Author Response · Authors · 2023-11-19
> **Response to the reviewer A2TK (#2)**
>
> ***
> ### **[W2]** (Robustness)
>
> To evaluate the robustness, we conducted three additional experiments on Cora, CiteSeer, and PubMed. For all experiments, we used GCN as the backbone model. We will include these results in the Appendix in our revised manuscript. We hope these experiments may resolve the reviewer’s concern.
>
> #### 1) Feature noise
>
> We evaluated the robustness when the feature noise is introduced in the graph. The feature noise is provided by adding standard Gaussian noise to input features with a probability of noise level. The below table shows how robust our method is for the feature noise. We indicated that our method outperforms the standard training in all noise levels.
>
>
> | Dataset   | Cora         |              |              | CiteSeer     |              |              | PubMed       |              |              |
> |-----------|--------------|--------------|--------------|--------------|--------------|--------------|--------------|--------------|--------------|
> | Noise level (%) | 10%          | 20%          | 40%          | 10%          | 20%          | 40%          | 10%          | 20%          | 40%          |
> | None      | 66.90 (1.54) | 66.89 (1.72) | 66.12 (1.73) | 78.10 (1.47) | 77.76 (1.54) | 77.92 (1.58) | 75.46 (1.67) | 74.62 (1.96) | 73.36 (2.49) |
> | iGraphMix | 68.48 (1.70) | 68.54 (1.58) | 68.30 (2.08) | 79.00 (2.41) | 78.64 (2.29) | 78.76 (2.94) | 75.78 (1.90) | 75.80 (2.03) | 74.88 (2.37) |
>
> #### 2) Structural noise
>
> We assessed the robustness when the graph contains structural noise. The structural noise is given by randomly adding and removing edges with a probability of noise level. The below table shows the results of how the proposed method is robust to the structural noise. We found that when structural noise is introduced, GCN trained with iGraphMix outperforms GCN trained without iGraphMix (None). In addition, as the noise level increases from 10% to 40%, our method achieves more performance improvement compared to the standard training (None).
>
> | Dataset   | Cora         |              |              | CiteSeer     |              |              | PubMed       |              |              |
> |-----------|--------------|--------------|--------------|--------------|--------------|--------------|--------------|--------------|--------------|
> | Noise level (%) | 10%          | 20%          | 40%          | 10%          | 20%          | 40%          | 10%          | 20%          | 40%          |
> | None      | 67.62 (1.02) | 61.88 (1.10) | 54.90 (1.97) | 78.62 (0.56) | 73.14 (0.37) | 60.96 (0.84) | 73.92 (0.37) | 71.88 (0.53) | 60.20 (0.58) |
> | iGraphMix | 68.94 (1.11) | 63.22 (0.38) | 55.82 (0.46) | 79.14 (0.47) | 74.06 (0.31) | 61.78 (0.82) | 74.54 (0.10) | 72.56 (0.54) | 61.22 (0.61) |
>
> #### 3) Label Noise
>
> We assessed the robustness when the graph contains label noise.  We randomly permuted training labels with a probability of noise level for this experiment.  We trained GCN with the noisy training labels and evaluated GCN's performance. The below table shows how robust our method for noise labels is. We indicated that our method outperforms the standard training in all noise levels.
>
> | Dataset   | Cora         |              |              | CiteSeer     |              |              | PubMed       |              |              |
> |-----------|--------------|--------------|--------------|--------------|--------------|--------------|--------------|--------------|--------------|
> | Noise level | 10%          | 20%          | 40%          | 10%          | 20%          | 40%          | 10%          | 20%          | 40%          |
> | None      | 79.10 (0.74) | 76.00 (1.07) | 67.60 (2.72) | 68.24 (2.40) | 64.14 (2.04) | 57.72 (4.65) | 77.32 (1.36) | 75.76 (2.52) | 68.64 (3.15) |
> | iGraphMix | 80.02 (0.86) | 76.42 (1.54) | 69.08 (2.11) | 69.08 (1.97) | 65.92 (2.30) | 58.62 (3.48) | 77.66 (1.55) | 76.00 (2.64) | 69.04 (3.59) |

---

> ### Author Response · Authors · 2023-11-19
> **Response to the reviewer A2TK (#3)**
>
> ***
> ### **[Q1]** (Time complexity)
>
> From Big O complexity analysis and the experiment, we indicated that our method does not exacerbate the computational speed. We will include the following result in the revised manuscript. Here is an in-depth analysis.
>
> We calculated the time complexity with the big O notation of all methods used in our paper. Let $\mathcal{V}$ be the number of nodes, $|\mathcal{E}|$ be the number of total edges, $|\mathcal{E}_{\mathrm{tr}}|$ be the number of edges connected to training nodes, $d$ be the hidden dimension of the GNN layer, and $K$ be the number of layers.
>
> Then, we can get the time complexity as shown in the below table. The time complexity of GCN was represented in the original GCN paper [1]. Since there are dropping edges or nodes in DropEdge and DropNode, the complexity for data preparation of DropEdge and DropNode is $\mathcal{O}(|\mathcal{E}|)$ and $\mathcal{O}(|\mathcal{E}|+\|\mathcal{V}\|)$, respectively. Also, the complexity of graph convolution is  $\mathcal{O}(p|\mathcal{E}|d^K)$, where $p \in [0, 1]$ is a dropping probability. DropMessage considers all edges and nodes in all layers, their complexity is the same as the standard GCN. For M-Mixup, the complexities for data preparation and graph convolution are $\mathcal{O}(\|\mathcal{E}\|+2\|\mathcal{V}\|)$ and  $\mathcal{O}(2|\mathcal{E}|d^K)$, respectively. Lastly, in the case of iGraphMix, the complexities for data preparation is $\mathcal{O}(\|\mathcal{E}\|+2\|\mathcal{V}\|)$ and that for graph convolution is $\mathcal{O}(\|\mathcal{E}\|d^K)$.
>
> | Operation           | None                              | DropEdge                           | DropNode                                       | DropMessage                        | M-Mixup                                         | iGraphMix (ours)                                |
> |---------------------|-----------------------------------|------------------------------------|------------------------------------------------|------------------------------------|-------------------------------------------------|-------------------------------------------------|
> | 1) Data Preparation | $\mathcal{O}(1)$                  | $\mathcal{O}(\|\mathcal{E}\|)$     | $\mathcal{O}(\|\mathcal{E}\|+\|\mathcal{V}\|)$ | $\mathcal{O}(1)$                   | $\mathcal{O}(\|\mathcal{E}\|+2\|\mathcal{V}\|)$ | $\mathcal{O}(\|\mathcal{E}\|+2\|\mathcal{V}\|)$ |
> | 2) Graph Convolution | $\mathcal{O}(\|\mathcal{E}\|d^K)$ | $\mathcal{O}(p\|\mathcal{E}\|d^K)$ | $\mathcal{O}(p\|\mathcal{E}\|d^K)$             | $\mathcal{O}(\|\mathcal{E}\|d^K)$ | $\mathcal{O}(2\|\mathcal{E}\|d^K)$             | $\mathcal{O}(\|\mathcal{E}\|d^K)$               |
> |                     |                                   |                                    |                                                |                                    |                                                 |                                                 |
>
> From this analysis, we found that our method would not need lots of computational complexity, especially compared to M-Mixup. Although some methods require less computational time, our method leads to better performance when training GNNs, highlighting the necessity of the proposed method.
>
> The below table shows the computational time of data preparation and graph convolution on Cora and ogbn-arxiv datasets. Due to the time constraint, only the computational time for Cora and ogbn-arxiv was represented in this rebuttal instead of covering all the data. The below table shows the mean computational time (second) per epoch. We could find that the real computational time is consistent with our time complexity calculation. We will report these results in the appendix section of the revised manuscript.
>
> | Dataset    | Operation           | Data Augmentation |           |           |         |           ||
> |------------|---------------------|------------|-----------|-----------|---------|-----------|----------|
> |            |                     | none       | DropEdge | DropNode |DropMessage|M-Mixup | iGraphMix |
> | Cora       | 1) Data preparation | **0.83e-3**    | 1.13e-3   | 1.33e-3   |**0.83e-3**| 1.96e-3 |  3.72e-3  |
> |            | 2) Graph Convolution         | 1.12e-2    | **1.10e-2**   | 1.10e-2   |1.11e-2| 1.73e-2 |  1.57e-2  |
> | ogbn-arxiv | 1) Data preparation | **1.25e-3**    | 1.34e-3   | 1.39e-3   | **1.25e-3** | 2.61e-2 |  2.56e-2  |
> |            | 2) Graph Convolution         | 3.32e-2    | **1.76e-2**   | 1.81e-2   | 3.30e-2 | 8.33e-2 |  3.37e-2  |
>
> ***
> [1] Kipf et al., Semi-supervised Classification with Graph Convolutional Networks, ICLR 2017

---

> ### Author Response · Authors · 2023-11-22
> **Reminder Comment**
>
> Dear the reviewer A2TK,
>
>
> We would like to appreciate your time and thank you for your valuable feedback. As the deadline for the Author/Reviewer discussion is approaching, we kindly ask if you could let us know whether our responses have addressed your concerns. This will significantly help us improve our work. If you require further clarification, please do not hesitate to reach out to us.
>
>
> Sincerely yours,
>
> Authors.

---

### Meta-Review · Area_Chair_HffJ · 2023-12-06

**Metareview:**

This paper introduces a new mixup method, iGraphMix, to address the irregularity and alignment issues for graph node classification. iGraphMix aggregates the sampled neighboring nodes instead of only interpolating node features. Moreover, the authors also provide theoretical analysis and show better generalization gap. Experimental results  demonstrate the effectiveness of iGraphMix.

Almost all reviewers agree with the novelty of the proposed method, and its sufficient experiments. Moreover, they also appreciate the theoretical analysis on the generalization performance of the proposed methods.  Only one reviewer who give negative score thinks that the compared baselines are simple, and not strong. The authors explain these strong methods need additional training techniques (e.g., min-max optimization, and robust optimization), and could potentially make a skewed comparison. For this point, I partially agree with the authors.  Consider the overall comments and the quality of this work, we accept this work.

**Justification For Why Not Higher Score:**

1) The average score of this work is 6.2, and is not very high.

2) One reviewer who give negative score thinks that the compared baselines are simple, and not strong.

**Justification For Why Not Lower Score:**

1) Almost all reviewers agree with the novelty of the proposed method, and its sufficient experiments. Moreover, they also appreciate the theoretical analysis on the generalization performance of the proposed methods.

2) Only one reviewer who give negative score thinks that the compared baselines are simple, and not strong. The authors explain these strong methods need additional training techniques (e.g., min-max optimization, and robust optimization), and could potentially make a skewed comparison. For this point, I partially agree with the authors.

---

### Decision · Program_Chairs · 2024-01-16

Accept (poster)